# Transport-related effects on intrinsic and synaptic properties of human cortical neurons: A comparative study

Guanxiao Qi[1] , Danqing Yang[1,2], Aniella Bak[3], Werner Hucko[1], Daniel Delev[4,5], Hussam Hamou[4], Dirk Feldmeyer[1,2,6] and Henner Koch[3]

[1] *Institute of Neuroscience and Medicine (INM-10), Research Centre Jüelich, Jülich, Germany*

[2] *Department of Psychiatry, Psychotherapy and Psychosomatics, RWTH Aachen University, Aachen, Germany*

[3] *Department of Epileptology and Neurology, RWTH Aachen Uniklinik, Aachen, Germany*

[4] *Department of Neurosurgery, RWTH Aachen Uniklinik, Aachen, Germany*

[5] *Department of Neurosurgery, University Hospital Erlangen, Friedrich-Alexander University, Erlangen Nürnberg, Erlangen, Germany*

[6] *Jülich-Aachen Research Alliance-Brain, Translational Brain Medicine, Aachen, Germany*

Handling Editors: Katalin Toth & Conny Kopp-Scheinpflug

The peer review history is available in the Supporting Information section of this article (https://doi.org/10.1113/JP288111#support-information-section).

**Guanxiao Qi** received his PhD in biology from RWTH Aachen University and is a research scientist at Research Centre Juelich. In collaboration with colleagues at University Hospital Aachen he studies the electrophysiology and morphology of neuronal microcircuits by single or paired patch-clamp recordings in acute human cortical slices prepared from neurosurgical resections and *post hoc* neuronal reconstruction. **Danqing Yang** received her PhD in neuroscience from RWTH Aachen University and is a postdoctoral researcher at Research Centre Juelich. By combining patch-clamp recordings, pharmacological manipulations and morphological reconstructions from human cortical neurons, she investigates how neuromodulators and disease-related genetic variants affect neuronal excitability and microcircuit function. **Aniella Bak** is a postdoctoral researcher in neuroscience at University Hospital Aachen, specializing in organotypic brain slice models of epilepsy and glioblastoma. Her work focuses on translational approaches in human brain tissue to reduce animal testing and improve preclinical modelling of neurological disorders.

G. Qi, D. Yang, and A. Bak shared the first authorship.

This article was first published as a preprint. Qi G, Yang D, Bak A, Hucko W, Delev D, Hamou H, Feldmeyer D, Koch H. 2024. Transport-Related Effects on Intrinsic and Synaptic Properties of Human Cortical Neurons: A Comparative Study. bioRxiv. https://doi.org/10.1101/2024.10.30.621044.

*The Journal of Physiology*

**Abstract figure legend** In this study, we investigated how transporting brain slices influences the electrophysiological and morphological properties of human cortical neurons. Whole-cell patch-clamp recordings were obtained from acute slices prepared from tissue collected during epilepsy or tumor surgeries, either on-site or off-site, after approximately 40 km of transport. Overall action potential (AP) firing patterns were largely preserved across recording locations, although some differences emerged. Pyramidal cells recorded off-site exhibited narrower APs and increased AP amplitudes compared to those recorded on-site. Fast-spiking (FS) interneurons showed a similar trend, with off-site recordings displaying narrower APs and higher AP amplitudes relative to their on-site counterparts. No differences in spontaneous excitatory postsynaptic potentials (sEPSP) frequency or amplitude were detected in pyramidal cells. In contrast, FS interneurons recorded off-site showed reduced sEPSP amplitude, whereas sEPSP frequency remained unchanged. Although overall dendritic architecture was preserved, spine densities in the apical oblique and apical tuft dendrites of off-site recorded pyramidal cells were reduced.

**Abstract** Transporting human brain tissue blocks or slices from the operating theatre or on-site laboratory to an off-site laboratory may affect sample integrity for electrophysiological studies.

In this study, we investigated how a 30–40 min transport influenced the intrinsic, synaptic and morphological properties of human cortical neurons. Electrophysiological recordings were performed on layer 2/3 (L2/3) pyramidal cells and fast-spiking (FS) interneurons from acute human cortical slices ($n = 200$ neurons from 32 surgeries, in which 112 neurons passed quality control for further analyses). Recordings were performed on-site at RWTH Aachen University Hospital and off-site at the Research Centre Jülich, which are approximately 40 km apart. Action potential (AP) firing patterns remained largely preserved across both recording sites, but several differences were observed. Off-site recorded pyramidal cells showed a depolarised resting membrane potential and a lowered rheobase current. In off-site recorded FS interneurons, we found a narrower AP half-width and an increased AP amplitude, suggesting altered ion channel kinetics and/or neuromodulatory environment. Additionally, a significant reduction in large rhythmic depolarisations and the amplitudes of spontaneous excitatory postsynaptic potentials in off-site recorded FS interneurons indicated an altered synaptic efficacy. Although overall dendritic architecture was preserved, the dendritic spine densities in apical oblique and apical tuft dendrites of off-site recorded pyramidal cells were also reduced. These findings emphasise the need for optimised transport conditions to preserve synaptic integrity, network activity and neuronal morphology. Standardised protocols are crucial for ensuring reliable and reproducible results in studies of human cortical function and structure.

(Received 15 November 2024; accepted after revision 8 December 2025; first published online 28 January 2026)

**Corresponding authors** D. Feldmeyer: Institute of Neuroscience and Medicine (INM-10), Research Centre Jülich, Leo-Brandt-Strasse 52425, Jülich, Germany.    Email: d.feldmeyer@fz-juelich.de, dfeldmeyer@ukaachen.de
Henner Koch: Department of Epileptology, Neurology RWTH Uniklinik Aachen, Pauwelsstraße, 3052074 Aachen, Germany.    Email: hkoch@ukaachen.de

## Key points

- *Effects of transportation on neuronal properties*: Brief transportation of human brain tissue retains many key neuronal properties, while still exhibiting measurable alterations in certain intrinsic, synaptic and morphological properties.
- *Mechanical stress and neuromodulator dysfunction may underlie alterations*: These changes are likely due to the combined effects of mechanical stress and altered neuromodulator signalling during transportation.
- *Advancing understanding of cortical function and structure*: This research provides valuable insights into the impact of transportation on human brain tissue, advancing our understanding of cortical function and structure and highlighting the importance of optimising transport protocols to preserve tissue integrity and neuronal function.

## Introduction

The investigation of human cortical microcircuits is attracting increasing interests within the neuroscience community, already spanning a broad range of studies focusing on cellular functions (Gidon et al., 2020; Peng et al., 2019; Verhoog et al., 2013), distinct cell types (Hodge et al., 2019; Lee et al., 2023; Yuste et al., 2020), the occurrence and modulation of network activity (Köhling et al., 1998; Yang et al., 2024) and the underlying mechanisms of both physiological and pathological conditions (Barth et al., 2021; Blauwblomme et al., 2019; Gorji et al., 2001; Köhling et al., 1999; Marcuccilli et al., 2010; Taylor et al., 2024). Given this growing interest, there

is a parallel need to optimise procedures for obtaining the highest quantity, best quality and longest survival of human brain samples, which is crucial for advancing research in this field (Bak et al., 2024; Kraus et al., 2020; Schwarz et al., 2017, 2019; Straehle et al., 2023; Wickham et al., 2018).

In *ex vivo* animal experiments, the location of sample collection and the site of experiments are generally in close proximity, that is, in the same laboratory, thereby largely eliminating the need to optimise sample transportation. In contrast, human tissue samples are obtained in the operating theatre of a clinic, and consequently, need to be transported from the surgery location to the

experimental laboratory. The duration of this transport can vary considerably, from minutes (Bak et al., 2024) to hours (Mittermaier et al., 2024), depending on the location of the experimental set-up.

Especially for longer distances, elaborate portable devices for blocks and/or slices of human brain samples were developed and tested before (Köhling et al., 1996; Mittermaier et al., 2024). Although these studies showed that brain samples can remain viable for successful recordings of action potential (AP) firing and synaptic activity in cortical neurons even after long-distance transport, more subtle changes may still occur during the transportation of the tissue.

In light of this, we aimed to systematically evaluate the impact of the transportation process on the intrinsic and synaptic properties of human cortical neurons. We compared two distinct settings: on-site analysis, where samples are rapidly transferred to a nearby laboratory in the same hospital, and off-site analysis, where samples undergo longer transport duration to a distant laboratory. By performing electrophysiological recordings in both scenarios, we aimed to assess potential differences in cellular properties, synaptic activity and overall neuronal viability. This comparative analysis provides critical insights into how transportation affects the quality and integrity of human brain samples, guiding future optimization of protocols for corresponding studies.

Our findings reveal that although firing patterns of human cortical neurons remain largely unchanged, significant alterations in several cellular parameters occur following the transport to an off-site laboratory. Pyramidal cells exhibited a more depolarised resting membrane potential and decreased rheobase current, along with decreased spine density when recorded off-site, and fast-spiking (FS) interneurons displayed an increased AP amplitude, decreased half-width and decreased amplitude of excitatory postsynaptic potentials (EPSPs). Furthermore, we observed a reduction in spontaneous network events recently described in human slices (Yang et al., 2024), known as large rhythmic depolarisations (LRDs), indicating that synaptic connectivity and activity are particularly susceptible to the effects of transport.

## Methods

### Ethical approval

Approval (EK-067/20) of the ethics committee of the University of Aachen, as well as written informed consent from all patients, was obtained to re-use spare, non-pathological brain tissue acquired during the surgical procedure for research in this study. The study conformed to the standards set by the Declaration of Helsinki, apart from registration in a database.

### Preparation of acute human brain slices

We collected data from the tissue samples of 32 patients (15 females, 17 males; age ranging from 9 to 75 years) (Table 1). In 17 patients, the indication for surgery was therapy-resistant epilepsy, whereas in 15 patients, the procedure was performed to remove a tumour. Tissue preparation was performed according to published protocols (Bak et al., 2024; Yang et al., 2024) (Fig. 1*A*). In brief, the dura mater and arachnoid were opened by the neurosurgeon with a scalpel and microscissors. The edges around the tissue were prepared with a dissector and ultrasonic tissue ablation. Finally, the arterial vessels to the tissue were closed and cut off with microscissors to minimize hypoxia as much as possible, and subsequently the tissue block was directly transferred into ice-cold ($\sim 4°C$) choline-based aCSF (in mM: 110 choline chloride, 26 $NaHCO_3$, 10 D-glucose, 11.6 Na-ascorbate, 7 $MgCl_2$, 3.1 Na-pyruvate, 2.5 KCl, 1.25 $NaH_2PO_4$ and 0.5 $CaCl_2$) equilibrated with carbogen (95% $O_2$, 5% $CO_2$) and immediately (within 5–10 min) transported to the laboratory. The tissue was always kept submerged in cooled and oxygenated slicing aCSF. Tissue blocks were primarily obtained from the temporal lobe ($n = 19$), with the remainder of the samples from the frontal ($n = 10$) and parietal ($n = 3$) lobes. After removal of the pia mater tissue blocks were trimmed perpendicular to the cortical surface and 300 μm-thick slices were prepared using a vibratome (Leica VT1200) in either the choline-based slicing aCSF or in ice-cold sucrose-based aCSF containing 206 mM sucrose, 2.5 mM KCl, 1.25 mM $NaH_2PO_4$, 3 mM $MgCl_2$, 1 mM $CaCl_2$, 25 mM $NaHCO_3$, 12 mM *N*-acetyl-L-cysteine and 25 mM glucose (325 mOsm/l, pH 7.45). During slicing, the solution was constantly bubbled with carbogen gas (95% $O_2$ and 5% $CO_2$). Subsequently, slices were first incubated for 30 min at 32–33°C and then at room temperature in aCSF containing (in mM) 125 NaCl, 2.5 KCl, 1.25 $NaH_2PO_4$, 1 $MgCl_2$, 2 $CaCl_2$, 25 $NaHCO_3$, 25 D-glucose (300 mOsm/l; 95% $O_2$ and 5% $CO_2$).

### Transportation of slices

To transport slices from the RWTH Aachen University Hospital to the Research Centre Jülich, a custom-made transportation system, including a custom-designed slice keeper and a portable carbogen gas bottle (Fig. 1*B*) or a 50 ml tube with saturated carbogen, was used. No significant differences in the intrinsic properties of off-site recorded neurons between the two transportation methods were found. The transportation time of the slices was between 30 and 40 min, covering approximately 40 km. During transportation, slices were kept at a temperature of 20–25°C in aCSF and constantly oxygenated with 95% $O_2$ and 5% $CO_2$. Upon arrival,

**Table 1. Patient details of brain surgery samples**

| Gender | Age | Diagnosis | Resection area | Transportation method | Preparation solution | Epilepsy (E) Tumour (T) |
|---|---|---|---|---|---|---|
| Female | 26 | Heterotopia | Temporal | No transport | Sucrose aCSF | E |
| Female | 35 | Cavernoma | Temporal | No transport | Sucrose aCSF | E |
| Female | 41 | Metastasis | Frontal | No transport | Sucrose aCSF | T |
| Female | 39 | Encephalitis | Parietal | No transport | Sucrose aCSF | E |
| Female | 75 | Glioma | Frontal | No transport | Sucrose aCSF | T |
| Male | 48 | Glioma | Frontal | No transport | Sucrose aCSF | T |
| Female | 62 | Cavernoma | Temporal | No transport | Sucrose aCSF | E |
| Male | 56 | Glioma | Frontal | No transport | Sucrose aCSF | T |
| Female | 69 | Metastasis | Frontal | No transport | Sucrose aCSF | T |
| Male | 17 | Glioma | Temporal | No transport | Sucrose aCSF | T |
| Female | 67 | Glioma | Parietal | No transport | Sucrose aCSF | T |
| Male | 9 | DNET | Temporal | No transport | Sucrose aCSF | E |
| Female | 41 | Hippocampus sclerosis | Temporal | Custom-made slice keeper | Sucrose aCSF | E |
| Female | 43 | Glioma | Temporal | Custom-made slice keeper | Sucrose aCSF | T |
| Male | 65 | Glioma | Temporal | Custom-made slice keeper | Sucrose aCSF | T |
| Male | 44 | Glioma | Frontal | Custom-made slice keeper | Sucrose aCSF | T |
| Male | 45 | Hippocampus sclerosis | Temporal | Custom-made slice keeper | Sucrose aCSF | E |
| Female | 74 | Meningioma | Temporal | Custom-made slice keeper | Sucrose aCSF | E |
| Male | 63 | Metastasis | Frontal | Custom-made slice keeper | Sucrose aCSF | T |
| Female | 48 | Hippocampus sclerosis | Temporal | Custom-made slice keeper | Sucrose aCSF | E |
| Male | 34 | DGONC | Fronto-basal | Custom-made slice keeper | Sucrose aCSF | E |
| Female | 37 | Hippocampus sclerosis | Temporal | Custom-made slice keeper | Sucrose aCSF | E |
| Male | 65 | Glioma | Temporal | Custom-made slice keeper | Sucrose aCSF | T |
| Male | 22 | A lesional epilepsy | Parieto-occipital | Custom-made slice keeper | Sucrose aCSF | E |
| Male | 32 | Hippocampus sclerosis | Temporal | Custom-made slice keeper | Sucrose aCSF | E |
| Male | 34 | Hippocampus sclerosis | Temporal | Custom-made slice keeper | Sucrose aCSF | E |
| Male | 47 | Hippocampus sclerosis | Temporal | Custom-made slice keeper | Choline aCSF | E |
| Female | 34 | Hippocampus sclerosis | Temporal | Custom-made slice keeper | Choline aCSF | E |
| Male | 55 | DNET WHO °I | Temporal | Oxygen-saturated aCSF | Sucrose aCSF | E |
| Male | 37 | Oligodendroglioma | Frontal | Oxygen-saturated aCSF | Sucrose aCSF | T |
| Female | 32 | Astrocytoma | Temporo-mesial | Oxygen-saturated aCSF | Sucrose aCSF | T |
| Male | 61 | Glioblastoma | Frontal | Oxygen-saturated aCSF | Choline aCSF | T |

*Note*: In this study tissue samples from $n = 32$ surgeries were used. Patient details on gender, age, diagnosis, resection area, transportation method, preparation solution and pathology of the tissue samples are given.

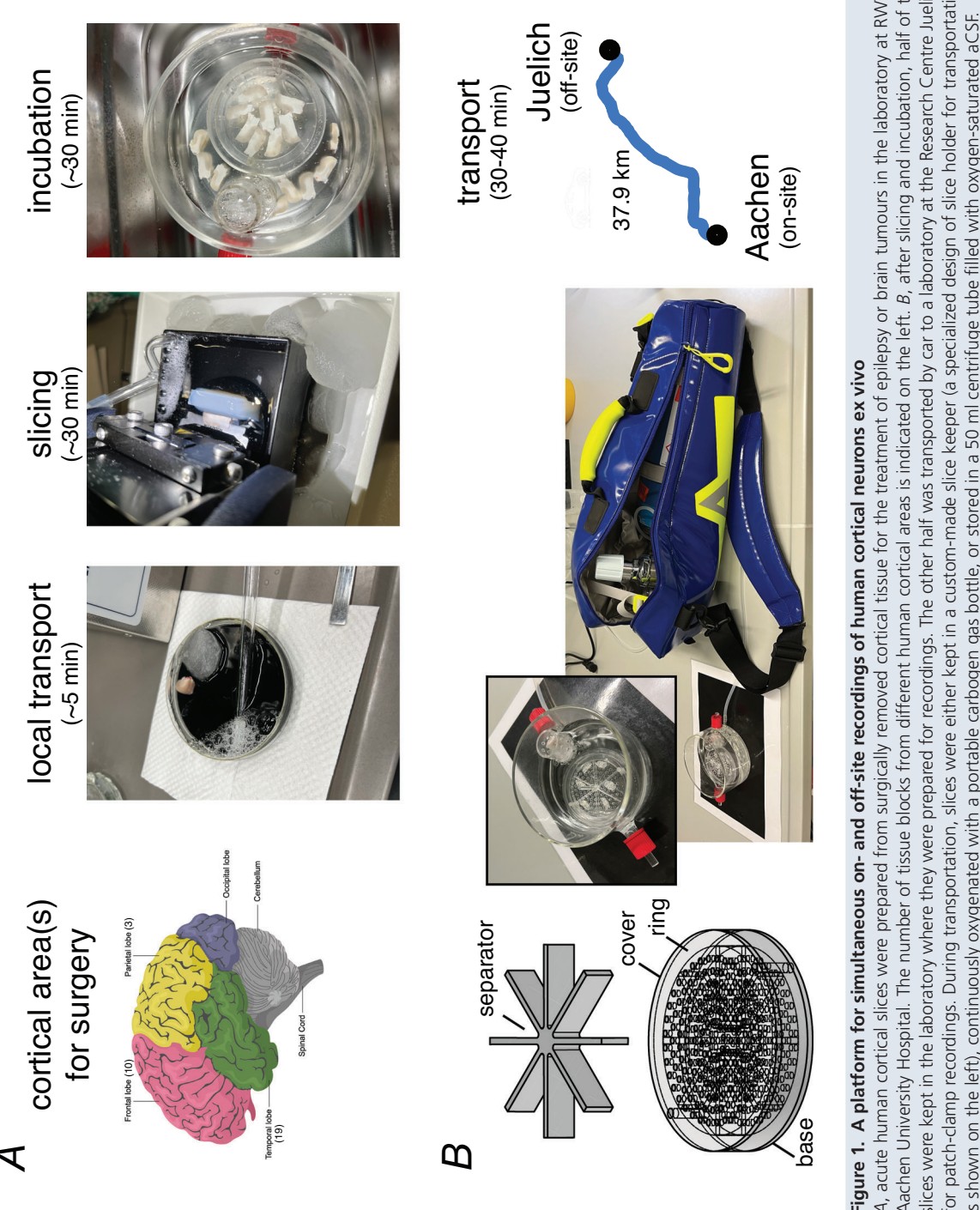

**Figure 1. A platform for simultaneous on- and off-site recordings of human cortical neurons ex vivo**

*A*, acute human cortical slices were prepared from surgically removed cortical tissue for the treatment of epilepsy or brain tumours in the laboratory at RWTH Aachen University Hospital. The number of tissue blocks from different human cortical areas is indicated on the left. *B*, after slicing and incubation, half of the slices were kept in the laboratory where they were prepared for recordings. The other half was transported by car to a laboratory at the Research Centre Juelich for patch-clamp recordings. During transportation, slices were either kept in a custom-made slice keeper (a specialized design of slice holder for transportation is shown on the left), continuously oxygenated with a portable carbogen gas bottle, or stored in a 50 ml centrifuge tube filled with oxygen-saturated aCSF.

**Table 2. Configurations of experimental set-ups and detailed information for patch-clamp recordings at on- and off-sites**

| List | On-site | Off-site |
|---|---|---|
| Amplifier | HEKA EPC 10 USB Double | HEKA EPC 10 USB Triple |
| Microscope | Zeiss Axio Examiner.D1 | Olympus BX51WI |
| Camera | TILL Photonics VX55 | TILL Photonics VX55 |
| Workstation | Luigs & Neumann Infrapatch 240 | Luigs & Neumann Infrapatch 240 |
| Faraday cage | Luigs & Neumann | Luigs & Neumann |
| Anti-vibration table | Newport Spectra-Physics | Newport Spectra-Physics |
| Micromanipulator | Luigs & Neumann SM-5 | Luigs & Neumann SM-5 |
| Micropipette puller | HEKA Sutter P-97 | HEKA Sutter P-1000 |
| Data acquisition software | HEKA Patchmaster | HEKA Patchmaster |
| Pipette resistance | 4−9 MΩ | 4−9 MΩ |
| Neuron selection | −50 to ∼ −100 μm | −50 to ∼ −100 μm |
| Recording temperature | 32−33°C | 32−33°C |
| Bridge balance of series resistance and capacitance | 80%, 10 μs | 80%, 10 μs |
| Experimenter | DY | GQ |

[Correction added on 17 February 2026 after first online publication: The experimenters listed for the on-site and off-site recordings were inadvertently reversed in the previous version and have now been corrected.]

slices were transferred into a bubble chamber filled with carbogenated aCSF identical to the on-site set-up in Aachen (Fig. 1).

### Whole-cell patch-clamp recordings

Whole-cell recordings were performed in acute slices on-site (RWTH Aachen University Hospital) and off-site (Research Centre Juelich). The set-ups used for on- and off-site recordings were nearly identical, with only minor differences, such as the light microscopy used for imaging (Table 2). Furthermore, the experimenters for on-site (DY) and off-site (GQ) recordings were trained in the same laboratory using the same patch-clamp set-up. These efforts were taken to reduce bias related to the set-up or the experimenter in the recording procedures. During the recordings, slices were continuously perfused (perfusion speed ∼5 ml/min) with aCSF, bubbled with carbogen gas and maintained at 32–33°C. Patch pipettes were pulled from thick-walled borosilicate glass capillaries and filled with an internal solution containing (in mM) 135 K-gluconate, 4 KCl, 10 HEPES, 10 phosphocreatine, 4 Mg-ATP and 0.3 GTP (pH 7.4, 290–300 mOsm/l). Whole-cell patch-clamp recordings were performed using patch pipettes with a resistance of 5–10 MΩ and obtained using an EPC10 amplifier (HEKA). Signals were sampled at 10 kHz, filtered at 2.9 kHz, with 80% series resistance and capacitance compensation using the Patchmaster software (HEKA), and later analysed offline using Igor Pro software (Wavemetrics). Biocytin was added to the internal solution at a concentration of 5 mg/ml for *post hoc* histology. Neurons were visualised using either Dodt gradient contrast microscopy on-site or infrared differential interference contrast (IR-DIC) microscopy off-site. Putative pyramidal cells and interneurons were differentiated based on the appearance of their cell bodies, their AP firing pattern during recordings and their morphological appearance after *post hoc* histological staining.

### Histological processing

After recordings, brain slices containing biocytin-filled neurons were fixed for at least 24 h at 4°C in 100 mM phosphate-buffered saline (PBS, pH 7.4) containing 4% paraformaldehyde (PFA), following a published protocol (Marx et al., 2012). After being rinsed several times in 100 mM PBS, slices were treated with 1% $H_2O_2$ in PBS for about 20 min to reduce any endogenous peroxidase activity. Slices were rinsed repeatedly with PBS and then incubated in 1% avidin-biotinylated horse-radish peroxidase (Vector ABC staining kit, Vector Lab. Inc.) containing 0.1% Triton X-100 for 1 h at room temperature. The reaction was catalysed using

0.5 mg/ml 3,3-diaminobenzidine (DAB; Sigma-Aldrich) as a chromogen. Subsequently, slices were rinsed with 100 mM PBS, followed by slow dehydration with ethanol in increasing concentrations, and finally in xylene for 2–4 h. Finally, slices were embedded using Eukitt medium (Otto Kindler GmbH).

### Morphological reconstructions

Using Neurolucida software (MBF Bioscience, Williston, VT, USA), morphological reconstructions of biocytin-filled human L2/3 pyramidal cells and FS interneurons were made at a magnification of 1000-fold (100-fold oil-immersion objective and 10-fold eyepiece) on an upright microscope. Neurons were selected for reconstruction based on the quality of biocytin labelling when background staining was minimal. Neurons with major truncations due to slicing were excluded. Embedding using Eukitt medium reduced the fading of cytoarchitectonic features and enhanced contrast between layers. This allowed the reconstruction of different layer borders along with the neuronal reconstructions. Furthermore, the position of soma and layers was confirmed by superimposing the Dodt gradient contrast or differential interference contrast images taken during the recording. The tissue shrinkage was corrected using correction factors of 1.1 in the $x$–$y$ direction and 2.1 in the $z$ direction (Marx et al., 2012). In addition to reconstructing dendritic branches, spines located on dendrites of pyramidal cells were counted. For spine counting, approximately 100 μm-length branches belonging to basal, apical oblique and apical tuft dendrites were selected for each pyramidal cell. Analysis of 3-D-reconstructed neurons was done using Neurolucida Explorer software.

### Data analysis

Custom-written macros for Igor Pro 6 (WaveMetrics) were used to analyse the recorded electrophysiological signals. The resting membrane potential ($V_{rest}$) of the neuron was measured directly after the breakthrough to establish the whole-cell configuration with zero current injection. Input resistance ($R_{in}$) was calculated as the slope of the linear fit to the current–voltage relationship. The membrane time constant ($\tau_m$) was determined by fitting a mono-exponential function to the initial portion of the hyperpolarising voltage response. Voltage sag ($V_{sag}$) was calculated as the difference between the initial trough of the hyperpolarized voltage and the steady-state voltage deflection divided by the steady-state deflection. For the analysis of single spike characteristics, such as threshold, amplitude, rise time, and half-width, a step size increment of 10 pA for current injection was applied to ensure that the AP was elicited very close to its rheobase current.

The spike threshold was defined as the point of start of acceleration of the membrane potential using the second derivative ($d^2V/dt^2$), that is, using $3\times$ standard deviation of $d^2V/dt^2$ as the cut-off point. The spike amplitude was calculated as the difference in voltage from the AP threshold to the peak during depolarisation. The spike rise time was calculated as the interval between the AP threshold and its peak. The spike half-width was determined as the time difference between the rising and decaying phases of the spike at half-maximum amplitude. Rheobase current was defined as the minimal current required to elicit the first AP. Afterhyperpolarisation (AHP) amplitude was calculated as the difference between the AP threshold and the trough of the fast AHP. To analyse repetitive firing properties, a series of current injections in 25 pA increments was applied to elicit trains of APs. Maximum firing frequency was defined as the greatest number of APs a neuron could generate during a 1 s depolarising current pulse. The frequency–current ($F$–$I$) slope was obtained from a linear fit to the $F$–$I$ response curve.

The spontaneous synaptic activity was analysed using the program SpAcAn (https://www.wavemetrics.com/project/SpAcAn). A 100 s segment of continuous, stable membrane-potential recording was analysed for pyramidal cells (PCs) because of their relatively low frequency of spontaneous EPSPs, whereas a 50 s segment was sufficient for interneurons owing to their higher EPSP frequency. In all cases, membrane-potential fluctuations during the selected window were <1 mV. Depending on the event type (EPSPs or LRDs) and the neuron type (pyramidal cell or FS interneuron), the number of events analysed ranged from 28 to 1452. EPSPs and LRDs were distinguished by differences in event amplitude and decay time (Yang et al., 2024). A threshold of 0.2 mV was set manually for detecting EPSP events, whereas a threshold of 3 mV was set for detecting LRDs. For both on- and off-site patch-clamp set-ups, noise levels were kept below 0.05 mV root mean square.

### Statistical analysis

Data are given as mean ± standard deviation (SD) and presented as box plots, in which the interquartile range (IQR) is shown as box, the range of values within 1.5*IQR is shown as whiskers and the median is represented by a horizontal line in the box. A Wilcoxon–Mann–Whitney $U$ test was performed to assess the difference between two samples. A Wilcoxon signed-rank test was performed to assess the difference between two paired samples. A $\chi^2$ test was performed to assess differences between proportions. In addition, we employed a linear mixed model to distinguish and estimate both fixed and random effects present in the dataset (Pinheiro & Bates, 1996).

Statistical significance was set at $P < 0.05$, and $n$ indicates the number of neurons analysed.

## Results

### Recordings at the off-site laboratory

In the first set of experiments, we investigated the viability of cortical neurons in slices for electrophysiological recordings after being transported to the off-site laboratory, as reported before (Mittermaier et al., 2024). We confirmed that human cortical neurons at the off-site laboratory remained viable for up to 60 h and showed only minor changes in passive membrane properties, for example, a decrease in $R_{in}$ but no significant change in their *post hoc* somatodendritic morphology (Fig. 2*A*) or firing properties (Fig. 2*B*) when compared between Day 1 and Day 3. However, we identified significant changes in spontaneous EPSP frequency and amplitude (Fig. 2*C*). The EPSP frequency significantly decreased in recordings obtained at Day 2 and Day 3 compared to Day 1. Furthermore, some paired recordings from synaptically coupled neurons were performed: in total one L2/3 pyramidal–pyramidal pair, one L2/3 pyramidal–interneuron pair (Fig. 3), one L2/3 interneuron–pyramidal pair and one L2/3 pyramidal–interneuron reciprocal pair were recorded from transported human cortical brain slices. These paired recordings demonstrated that synaptically connected neurons can still be recorded after tissue transportation, as previously reported (Mittermaier et al., 2024).

### Off-site recordings show different intrinsic properties after transportation

To determine whether acute human brain slices can be reliably used for electrophysiological patch-clamp recordings in a laboratory located off-site from the clinic where the slices were prepared, and to assess if transport affects the properties of recorded neurons, we performed either on-site or off-site recordings or parallel on- and off-site recordings ($n = 32$ surgeries, see Table 1). Slices were recorded on the day of surgery (Day 1) either at the RWTH Aachen University Hospital (on-site) or at the Research Centre Juelich (off-site). There is an approximate 1.5 h delay between on- and off-site recordings due to slice transportation and other handling. Note that comparisons between on- and off-site electrophysiological recordings were conducted within the first 12 h after surgery.

Of the 200 recorded neurons, 133 were recorded on-site and 67 were recorded off-site. First, we compared the intrinsic firing properties of L2/3 pyramidal cells and FS interneurons recorded in parallel on- and off-sites.

Only neurons with a stable $V_{rest}$ below $-60$ mV and a series resistance smaller than 50 M$\Omega$ were included in the analysis to ensure a high recording quality.

L2/3 pyramidal cells ($n = 25$ on-site and $n = 20$ off-site) at both recording sites exhibited the characteristic firing pattern of regular AP firing with a spike frequency adaptation (Fig. 4*A*), as previously described (Avoli et al., 1994). However, several intrinsic properties of these neurons showed significant differences between on- and off-site recordings (Fig. 4*B*). Specifically, pyramidal cells recorded at the off-site laboratory were slightly more depolarised, exhibiting a less negative $V_{rest}$ compared to those recorded on-site. Additionally, off-site pyramidal cells had a significantly lower rheobase current and a higher maximal firing frequency compared to their on-site counterparts under the same stimulation amplitude (Table 3). In conclusion, neurons recorded off-site were more excitable and required less current input to reach the threshold for AP generation.

A linear mixed effects (LMEs) model was used to analyse neuronal intrinsic properties of L2/3 pyramidal cells. We used the location of the recordings (on-site *vs.* off-site) as the fixed effect and the surgery as a random effect to account for potential clustering within surgical sessions. The analysis revealed that 5 out of 12 parameters showed statistically significant differences between on- and off-sites: $V_{rest}$ ($P = 1.4 \times 10^{-5}$), time constant ($P = 7.5 \times 10^{-4}$), rheobase current ($P = 0.004$), AP amplitude ($P = 0.01$) and maximum firing frequency ($P = 1.5 \times 10^{-6}$), whereas other parameters, for example, the $R_{in}$ ($P = 0.53$), sag ($P = 0.14$), AP threshold ($P = 0.60$), AP rise time ($P = 0.30$), AP half-width ($P = 0.22$), AHP amplitude ($P = 0.092$), *F–I* slope ($P = 0.093$), showed no significant differences. Compared to the statistical results obtained with the non-parametric Wilcoxon–Mann–Whitney two-sample rank test (see Fig. 4 and Table 3), the analysis using the LME model showed similar results except for the AP half-width. Only $V_{rest}$ and *F–I* slope showed a meaningful between-surgery variability (intraclass correlation coefficient, ICC = 0.293 and 0.159, respectively). This indicates that the surgical cases contributed to the overall variance in these two parameters. The other analysed values showed minimal variance between the surgeries (ICC $\approx$ 0), suggesting that the primary source of variation was at the individual recording rather than between surgical sessions.

Patch-clamp recordings from human cortical neurons were not always successful at both on- and off-site locations for each slice transportation. To enable paired comparisons, parallel on- and off-site recordings were obtained from six surgeries, yielding 20 pyramidal cells recorded on-site and 16 off-site. Electrophysiological parameters for comparison were determined by averaging the values from individual pyramidal cells recorded from the same surgery. Paired comparison revealed significant

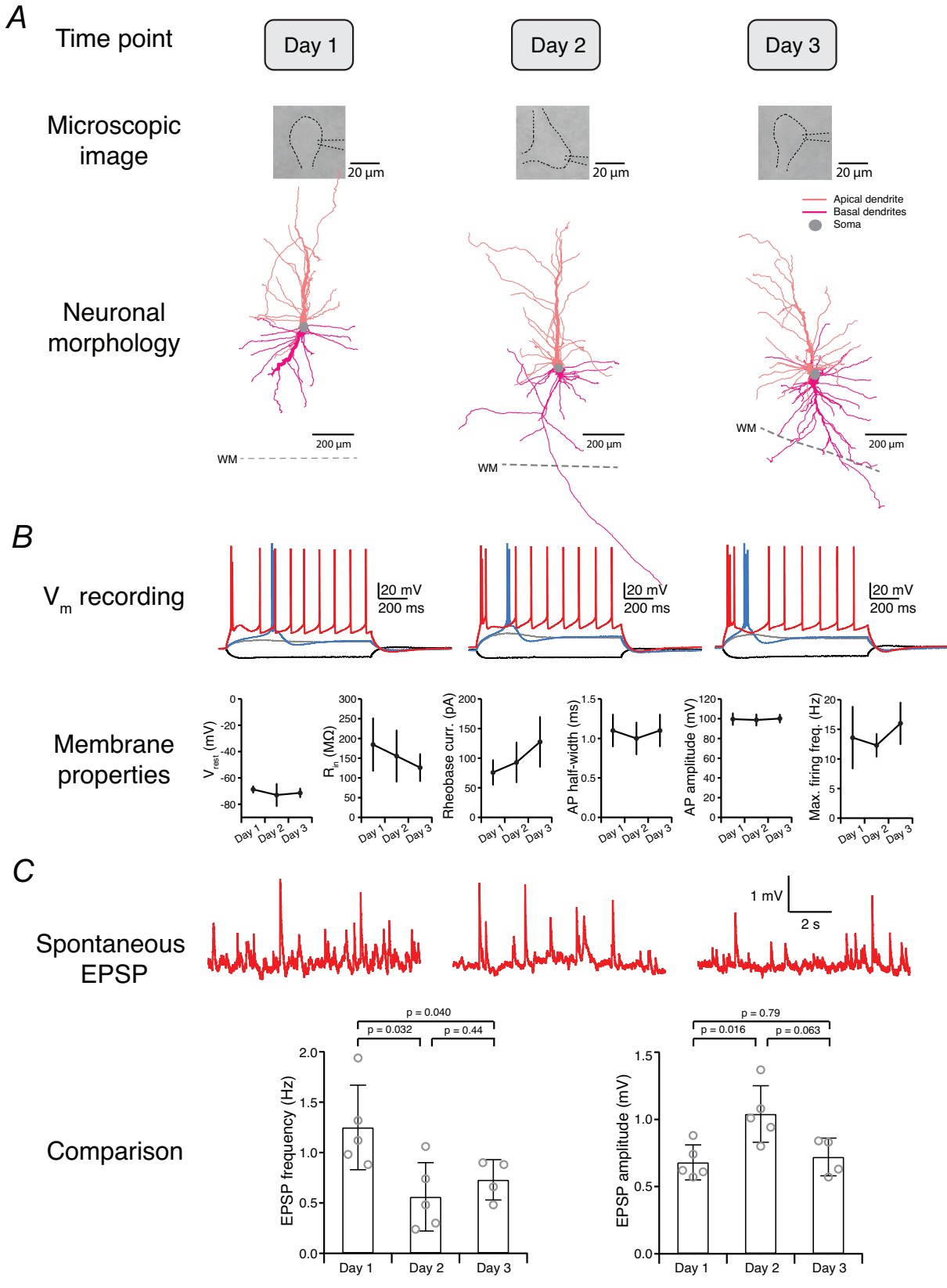

**Figure 2. Prolonged (∼ 60 h) survival of human cortical neurons maintained *ex vivo* after transportation**
*A*, representative patch-clamp recordings from cortical neurons in *ex vivo* slices on Day 1, Day 2 and Day 3 after slice preparation. Bottom row shows morphological reconstructions of three recorded human cortical neurons,

of which $V_m$ recordings are shown below. Cell bodies of these neurons were located in deep cortical layers near the white matter (WM). B, current-clamp recordings show the AP firing response to current injections. Bottom panel shows changes in intrinsic membrane properties over 3 days of recordings; n = 5 for Day 1, n = 7 for Day 2 and n = 4 for Day 3. C, representative excitatory postsynaptic potential (EPSP) recordings from cortical neurons maintained in *ex vivo* at three time points after the slice preparation. Bottom panel shows the comparison of EPSP frequency and amplitude at three time points after the slice preparation. *P*-values were calculated using the Wilcoxon–Mann–Whitney two-sample rank test.

differences in $V_{rest}$ ($-77.6 \pm 5.2$ mV, $n = 6$ on-site *vs.* $-66.7 \pm 5.1$ mV, $n = 6$ off-site; $P = 0.031$) and maximal firing frequency ($17.5 \pm 1.5$ Hz, $n = 6$ on-site *vs.* $29.3 \pm 3.7$ Hz, $n = 6$ off-site; $P = 0.031$). Other parameters, such as rheobase current ($106.0 \pm 18.1$ pA, $n = 6$ on-site *vs.* $58.9 \pm 19.8$ pA, $n = 6$ off-site; $P = 0.094$) and AP amplitude ($90.7 \pm 1.9$ mV, $n = 6$ on-site *vs.* $97.1 \pm 6.0$ mV, $n = 6$ off-site; $P = 0.094$), also showed differences, which were, however, statistically not significant.

An objective comparison of on- and off-site data may be influenced by systematic bias introduced by using different set-ups or different experimenters. To minimise the potential for such bias, we used experimental set-ups at both sites that were very similar in composition (see Table 2). Additionally, the experimenters who conducted patch-clamp recordings at both on- and off-site locations had very similar technical expertise. To further minimise bias related to set-ups and experimenters, all

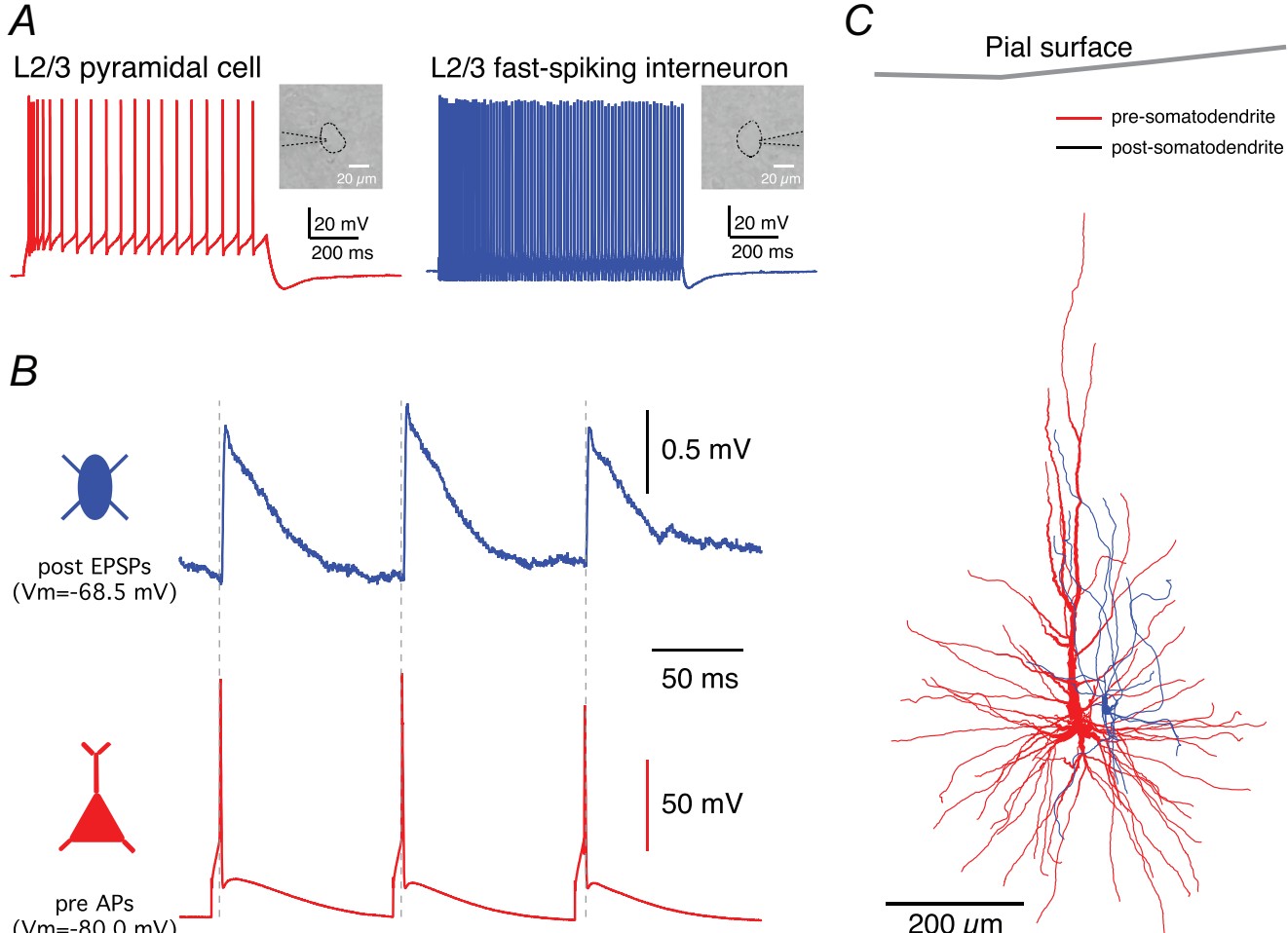

**Figure 3. Paired recordings from synaptically coupled neurons in acute human cortical slices directly after transportation**

A, firing patterns of pre- and postsynaptic neurons for a synaptically coupled human L2/3 pyramidal–fast-spiking interneuron pair recorded in *ex vivo* slices after transportation. Inserts: DIC images of recorded neurons. B, postsynaptic EPSPs (top) were evoked by presynaptic APs (bottom). Note the time lock between individual presynaptic APs and postsynaptic EPSPs. C, morphological reconstructions of the somatodendritic domain of pre- and postsynaptic neurons based on the *post hoc* biocytin staining.

experiments were performed by the same experimenter using human cortical slices obtained from three surgeries. These experiments were conducted either directly in the on-site laboratory or after a 30–40 min round-trip transport.

Comparison of transported *vs.* non-transported slices revealed statistically significant differences in $V_{rest}$ ($-74.3 \pm 2.0$ mV, $n = 7$ for transported *vs.* $-78.6 \pm 3.5$ mV, $n = 7$ for non-transported; $P = 0.026$), AP half-width ($0.78 \pm 0.12$ ms, $n = 7$ for transported *vs.* $0.92 \pm 0.13$ ms, $n = 7$ for non-transported; $P = 0.026$) and AP amplitude ($91.4 \pm 4.2$ mV, $n = 7$ for transported *vs.* $85.0 \pm 7.8$ mV, $n = 7$ for non-transported; $P = 0.038$), consistent with the results shown in Fig. 4*B*. In contrast no significant differences were observed in spontaneous EPSP frequency ($3.7 \pm 1.4$ Hz, $n = 7$ *vs.* $4.1 \pm 1.7$ Hz, $n = 7$; $P = 0.71$) and amplitude ($0.39 \pm 0.10$ mV, $n = 7$ *vs.* $0.47 \pm 0.20$ mV, $n = 7$; $P = 0.30$), consistent with the results shown in Fig. 6*B*.

These results suggest that although the overall firing capability of L2/3 pyramidal cells is independent of the recording location, specific intrinsic properties, such as $V_{rest}$, AP half-width and AP amplitude, are influenced by the transportation process. Similarly, FS interneurons ($n = 14$ on-site and $n = 13$ off-site) recorded at both locations demonstrated the expected characteristic high-frequency firing with narrow APs and no spike frequency adaptation (Fig. 5*A*). However, significant differences were observed in specific AP parameters between the on- and off-site recordings (Fig. 5*B* and Table 3).

Although the $V_{rest}$ was comparable between FS interneurons recorded on- and off-site, the AP half-width was narrower in interneurons recorded off-site compared to those recorded on-site, indicating a change in the temporal dynamics of AP generation (see Table 3). Additionally, the amplitude of the APs was significantly increased in off-site recorded FS interneurons. These findings indicate that transport to an off-site laboratory may affect the biophysical properties of FS interneurons, potentially affecting their roles in network dynamics and inhibitory control within the cortical circuitry.

For consistency, we also used an LME model to analyse the intrinsic properties of L2/3 FS interneurons. The analysis revealed that only 2 out of 12 parameters showed

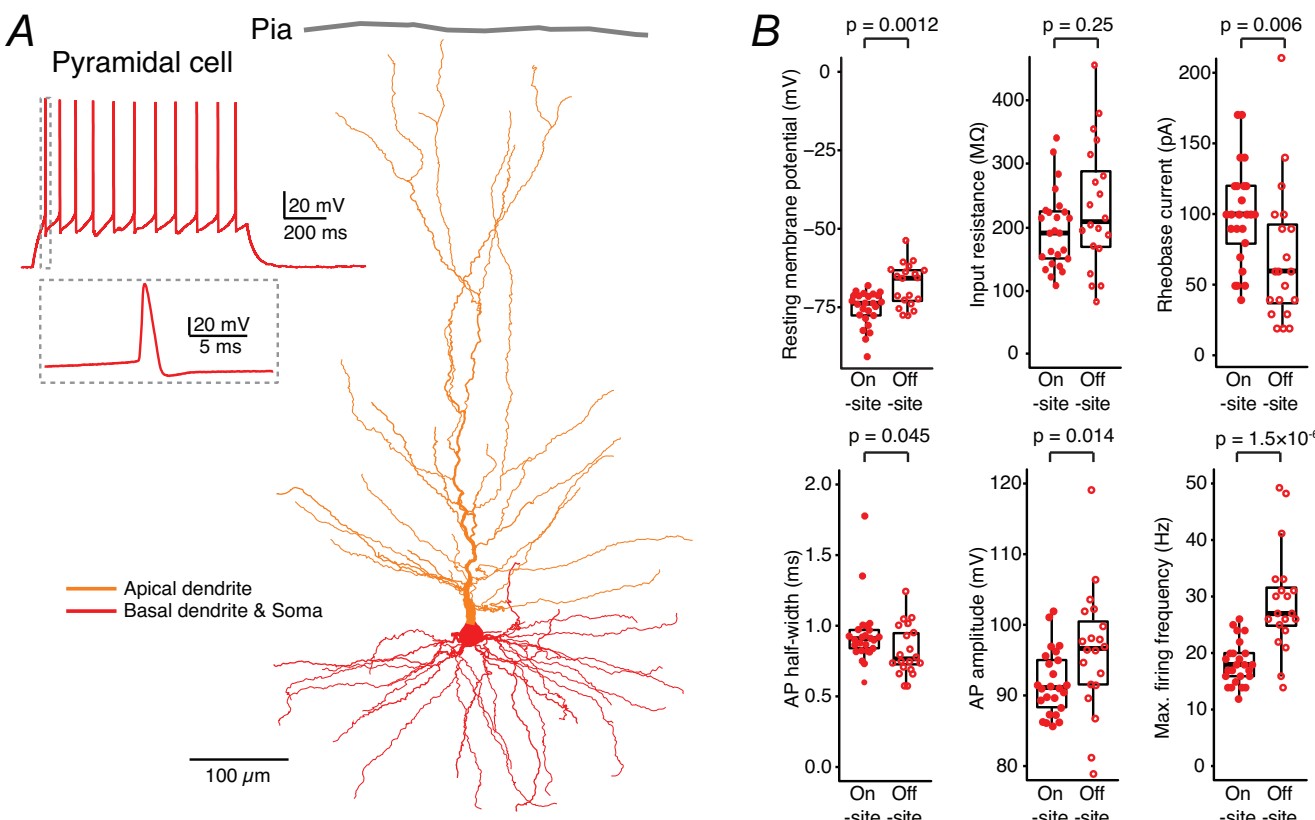

**Figure 4. Comparison of the electrophysiological properties between on- and off-site recorded human cortical L2/3 pyramidal cells**
*A*, typical firing pattern (regular spiking) and the morphological reconstruction of a human cortical L2/3 pyramidal cell. *B*, comparison of electrophysiological properties of pyramidal cells recorded on-site ($n = 25$ from 8 surgeries) and off-site ($n = 20$ from 9 surgeries). *P*-values were calculated using the Wilcoxon–Mann–Whitney two-sample rank test.

**Table 3. Comparison of electrophysiological properties between on- and off-site recorded human cortical neurons**

| | L2/3 pyramidal cell | | | L2-3 fast-spiking interneuron | | |
|---|---|---|---|---|---|---|
| | On-site ($n = 25$) | Off-site ($n = 20$) | $P$-value | On-site ($n = 14$) | Off-site ($n = 13$) | $P$-value |
| ***Passive*** | | | | | | |
| $V_{rest}$ (mV) | $-75.2 \pm 5.5$ | $-67.6 \pm 6.7$ | $1.2 \times 10^{-3}$ | $-64.9 \pm 6.2$ | $-65.2 \pm 4.2$ | 0.94 |
| Rin (MΩ) | $192.6 \pm 62.2$ | $230.0 \pm 101.9$ | 0.25 | $453.7 \pm 401.3$ | $239.6 \pm 163.9$ | 0.094 |
| $\tau_m$ (ms) | $18.4 \pm 4.3$ | $24.7 \pm 7.6$ | $2.3 \times 10^{-3}$ | $12.6 \pm 5.0$ | $10.1 \pm 2.4$ | 0.35 |
| Sag (%) | $8.5 \pm 9.5$ | $11.8 \pm 11.9$ | 0.41 | $21.9 \pm 16.0$ | $16.5 \pm 7.6$ | 0.55 |
| ***Single AP*** | | | | | | |
| Rheobase current (pA) | $99.2 \pm 34.8$ | $69.5 \pm 48.0$ | $6.0 \times 10^{-3}$ | $50.0 \pm 45.4$ | $68.5 \pm 51.5$ | 0.26 |
| AP threshold (mV) | $-41.2 \pm 3.9$ | $-40.2 \pm 3.2$ | 0.22 | $-41.6 \pm 4.6$ | $-40.7 \pm 3.7$ | 0.83 |
| AP rise time (ms) | $0.51 \pm 0.10$ | $0.48 \pm 0.06$ | 0.42 | $0.36 \pm 0.08$ | $0.31 \pm 0.04$ | 0.11 |
| AP half-width (ms) | $0.95 \pm 0.21$ | $0.83 \pm 0.18$ | 0.045 | $0.32 \pm 0.08$ | $0.23 \pm 0.04$ | $2.6 \times 10^{-4}$ |
| AP amplitude (mV) | $91.8 \pm 4.5$ | $95.9 \pm 9.4$ | 0.014 | $82.0 \pm 7.4$ | $97.9 \pm 9.7$ | $2.0 \times 10^{-4}$ |
| AHP amplitude (mV) | $15.2 \pm 4.0$ | $13.3 \pm 2.6$ | 0.21 | $24.7 \pm 4.7$ | $26.8 \pm 2.8$ | 0.12 |
| ***Repetitive firing*** | | | | | | |
| Maximum firing frequency (Hz) | $18.2 \pm 3.8$ | $29.0 \pm 9.0$ | $1.5 \times 10^{-6}$ | $116.9 \pm 29.7$ | $122.7 \pm 52.9$ | 0.95 |
| *F–I* slope (APs/100 pA) | $18.0 \pm 6.7$ | $14.9 \pm 4.4$ | 0.20 | $58.1 \pm 31.7$ | $46.1 \pm 16.4$ | 0.35 |

*Note*: *P*-value was calculated using the non-parametric Wilcoxon–Mann–Whitney two-sample rank test. Data are presented as mean $\pm$ SD.

statistically significant differences between on- and off-site recordings: AP half-width ($P = 5.5 \times 10^{-4}$) and AP amplitude ($P = 3.3 \times 10^{-5}$), whereas other parameters, for example, the $V_{rest}$ ($P = 0.32$), $R_{in}$ ($P = 0.059$), time constant ($P = 0.23$), sag ($P = 0.11$), rheobase current ($P = 0.41$), AP threshold ($P = 0.47$), AP rise time ($P = 0.065$), AHP amplitude ($P = 0.18$), maximum firing frequency ($P = 0.71$) and *F–I* slope ($P = 0.51$), showed no significant differences. Compared to the statistical results obtained with the non-parametric Wilcoxon–Mann–Whitney two-sample rank test (see Fig. 5 and Table 3), the analysis using the LME model gave very similar results. As data at the paired surgery level or for transport-only conditions were unavailable for FS interneurons, no additional analyses were conducted for these scenarios.

## Comparison of spontaneous EPSP frequency and amplitude between on- and off-site recorded human cortical neurons

To investigate whether the transport of human cortical slices to an off-site laboratory affects synaptic properties, we compared the frequency and amplitude of spontaneously generated EPSPs recorded in L2/3 pyramidal cells and FS interneurons at on- and off-sites. The majority of the neurons included in this analysis were derived from surgeries where recordings were performed both on- and off-site. L2/3 pyramidal cells recorded both on-site (Fig. 6*A* upper trace) and off-site (Fig. 6*A* lower trace) exhibited spontaneous EPSPs (sEPSPs) (Fig. 6*A*, left).

Quantitative analysis confirmed that there were no significant differences in the frequency or amplitude of sEPSPs between pyramidal cells recorded on-site ($n = 12$) and those recorded off-site ($n = 12$) (Fig. 6*B*). Similarly analysis of L2/3 FS interneurons recorded on-site ($n = 12$) and off-site ($n = 12$) revealed no significant differences in sEPSP frequency between the two groups ($6.5 \pm 5.5$ Hz on-site and $5.0 \pm 4.1$ Hz off-site; $P = 0.84$) (Fig. 6*C* and 6*D*, left). However, a significant decrease in sEPSP amplitude was observed in interneurons recorded off-site compared to those recorded on-site ($1.02 \pm 0.33$ mV *vs.* $2.05 \pm 1.08$ mV; $P = 0.0058$) (Fig. 6*D*, right). This reduction in sEPSP amplitude suggests that the synaptic efficacy of excitatory inputs onto FS interneurons is particularly affected by transportation. Overall, these findings indicate that although the frequency of sEPSPs was similar between on- and off-site recordings for both pyramidal cells and FS interneurons, the amplitude of sEPSPs in FS interneurons was significantly reduced when recordings were conducted off-site compared to on-site.

This suggests that transport conditions may affect synaptic transmission, particularly in excitatory-to-inhibitory circuits. This, in turn, may affect

the capacity of these circuits to modulate and generate network activity.

## Comparison of LRD occurrence in brain slice samples recorded on- and off-site

Next, we recorded the spontaneous activity of neurons in current-clamp mode to determine the presence of spontaneously generated LRDs, which we have described recently (Yang et al., 2024). At the on-site laboratory, 10% of the recorded neurons ($n = 133$, including 64 pyramidal cells and 69 interneurons) exhibited spontaneous LRDs, whereas only 3% of the neurons recorded at the off-site laboratory ($n = 67$, including 29 pyramidal cells and 38 interneurons) displayed spontaneous LRDs (sLRDs) (Fig. 7*A–C*). There exists a significant difference in the percentage of on- and off-site recorded neurons displaying sLRDs ($P = 0.043$ for $\chi^2$ test without Yates correction). Among the 13 neurons recorded on-site that exhibited sLRDs, 5 were FS interneurons, 5 were non-FS (nFS) interneurons and 3 were pyramidal cells. In contrast,

both neurons recorded off-site that exhibited sLRDs were FS interneurons. Interestingly, neurons displaying sLRDs that were recorded at the off-site laboratory were located deeper in the slice than those neurons showing no sLRDs: the depths of soma to the slice surface was 85 and 100 μm with a mean $\pm$ SD of 92.5 $\pm$ 10.6 μm ($n = 2$) for sLRD+ neurons and 73.5 $\pm$ 15.7 μm ($n = 30$) for sLRD− neurons in which the somatic depths were measured during recordings. Additionally, in a subset of neurons, we used norepinephrine (NE, 10 or 30 μM) as a pharmacological stimulant to induce LRDs, defined as the induced LRDs (iLRDs) (Yang et al., 2024). We found that 21% of the neurons recorded at the on-site laboratory responded with LRDs following NE application, whereas only 7% of the neurons recorded off-site exhibited a similar response (Fig. 7*D, E*). Therefore, the incidence of LRDs was significantly reduced at the off-site laboratory (off-site 7% *vs.* on-site 21%; $P = 0.031$ for $\chi^2$ test without Yates correction), both in the absence and presence of NE-induced modulation. In addition to the incidence of LRDs between on- and off-site recordings, the amplitude and frequency of LRDs were analysed. No significant

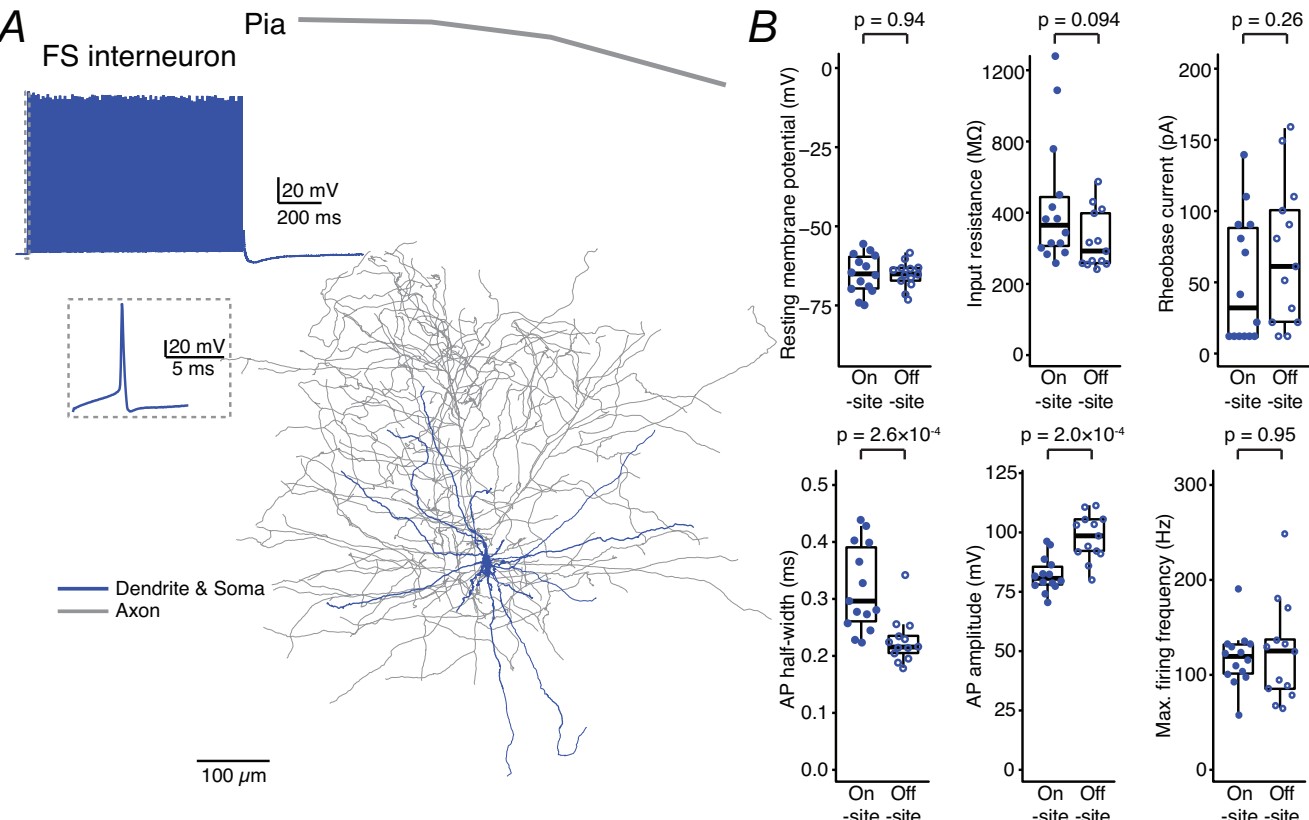

**Figure 5. Comparison of electrophysiological properties of human fast-spiking interneurons recorded on- and off-site**
*A*, representative firing pattern and morphological reconstruction of a human cortical L2/3 fast-spiking interneuron. *B*, comparison of electrophysiological properties of FS interneurons recorded on-site ($n = 14$ from 10 surgeries) and off-site ($n = 13$ from 9 surgeries). *P*-values were calculated using the Wilcoxon–Mann–Whitney two-sample rank test.

differences were observed in amplitude (9.0 ± 3.5 mV, $n = 20$ *vs*. 6.4 ± 2.7 mV, $n = 5$; $P = 0.15$) or frequency (0.31 ± 0.18 Hz, $n = 20$ *vs*. 0.34 ± 0.17 Hz, $n = 5$; $P = 0.61$) between on- and off-site recorded LRDs. Similarly, no significant differences were observed in amplitude (8.7 ± 4.2 mV, $n = 13$ *vs*. 9.6 ± 1.4 mV, $n = 7$; $P = 0.39$) or frequency (0.35 ± 0.19 Hz, $n = 13$ *vs*. 0.24 ± 0.15 Hz, $n = 7$; $P = 0.22$) when comparing sLRDs to NE-induced LRDs.

In summary, based on experiments with NE, we conclude that altered neuromodulator signalling during slice transport and handling likely contributes to the reduction in LRDs.

## Morphological analysis of pyramidal cells and FS interneurons

The *post hoc* morphological analysis of pyramidal cells and FS interneurons recorded either on-site or off-site revealed no significant differences between the two groups in overall dendritic structure, including dendritic length and branching pattern (Fig. 8 and Table 4). This indicates that despite the observed differences in intrinsic electrophysiological membrane properties and synaptic properties, the transport of brain slices to an off-site laboratory does not affect the overall structural integrity of the neurons.

However, a detailed analysis of the dendritic spine density of L2/3 pyramidal cells (Fig. 9*A*) revealed significant differences between the two recording sites. Specifically, we found that the spine densities in the apical oblique and apical tuft dendrites were significantly reduced in pyramidal cells recorded off-site compared to those recorded on-site (Fig. 9*B*).

In contrast, no significant differences were observed in the basal dendritic compartments of the two groups. These findings suggest that although the overall dendritic (and likely also axonal) architecture remains intact, the

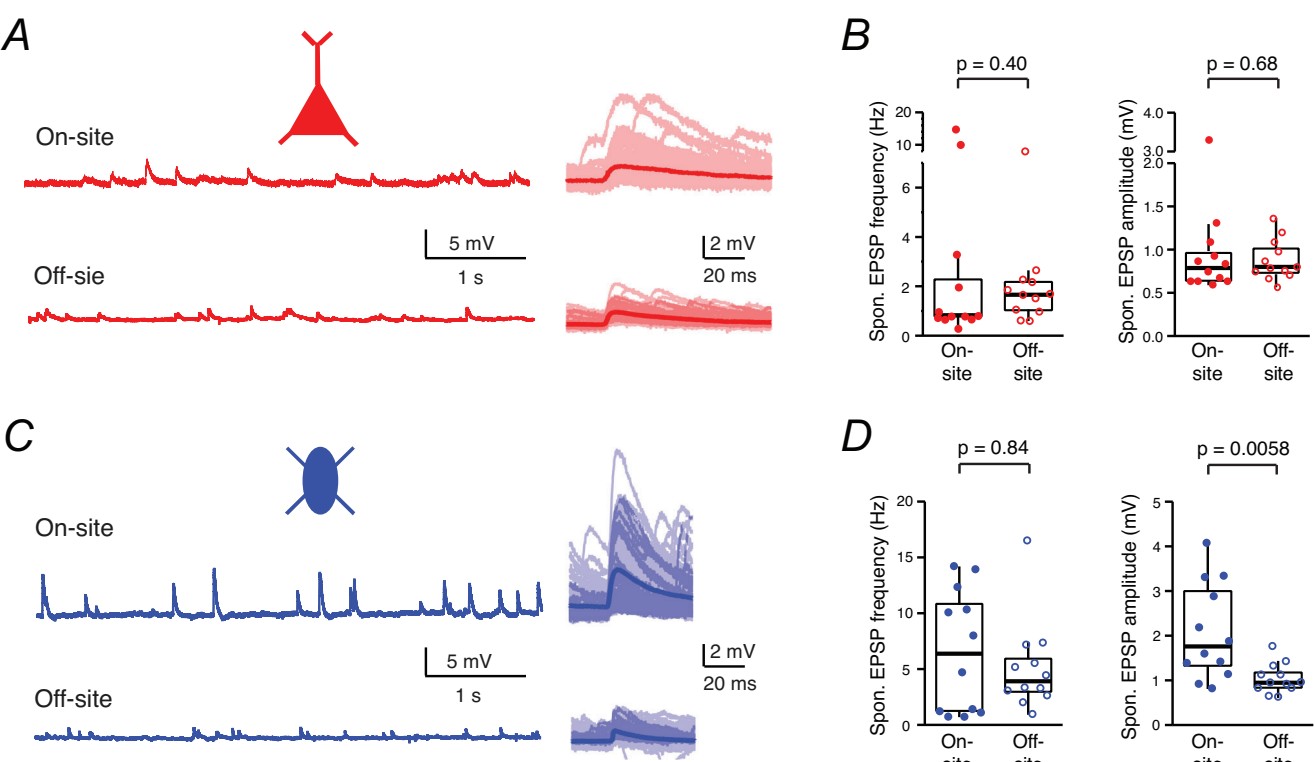

**Figure 6. Comparison of spontaneous excitatory postsynaptic potential (EPSP) frequency and amplitude between on- and off-site recorded human cortical neurons**

*A*, left: representative $V_m$ traces of human cortical L2/3 pyramidal cells recorded on-site (upper trace) and off-site (lower trace). Right: overlay of single spontaneous EPSPs (sEPSPs) and their average for visual comparison. *B*, comparison of sEPSP frequency (left) and amplitude (right) in pyramidal cells recorded on-site ($n = 12$ from 5 surgeries) *vs*. off-site ($n = 12$ from 7 surgeries). No significant differences were observed in either EPSP frequency or amplitude between the two locations. *C*, *D*, similar analysis as in *A* and *B*, but for human cortical L2/3 fast-spiking interneurons recorded on-site ($n = 12$ from 6 surgeries) and off-site ($n = 12$ from 10 surgeries). Although there were no significant differences in sEPSP frequency between on- and off-site recordings, a significant decrease in sEPSP amplitudes was observed in off-site recorded FS interneurons compared to those recorded on-site. *P*-values were calculated using the Wilcoxon–Mann–Whitney two-sample rank test.

finer synaptic structures, such as dendritic spines, are more susceptible to transport-related effects, potentially contributing to the observed alterations in synaptic efficacy and network activity.

## Discussion

This study provides a systematic analysis of the effects of transportation on the intrinsic and synaptic properties of human cortical neurons by comparing on- and off-site recordings. Although a successful transportation of human brain tissue for electrophysiological recordings has been previously documented (Andersson et al., 2016; Köhling et al., 1996; Mittermaier et al., 2024), detailed investigations into subtle changes in morphological, electrophysiological and network properties are required to assess whether transport affects tissue integrity and neuronal function. Our results indicate that neurons in brain slices transported for 30–40 min from the surgical resection site to an off-site laboratory retain notable robustness of many key electrophysiological (e.g. AP kinetics, sEPSP activity) and morphological

(e.g. overall dendritic architecture) properties, while still exhibiting measurable alterations in certain intrinsic, synaptic and morphological properties. These alterations are cell type-specific, for example, $V_{rest}$ changes only occur in pyramidal cells and sEPSP amplitude changes only occur in FS interneurons.

### Alterations in the intrinsic properties of neurons

Our data show that although the overall firing capabilities of L2/3 pyramidal cells and FS interneurons are preserved between on- and off-site recordings, there are subtle yet significant differences in intrinsic properties. Off-site recorded pyramidal cells were more depolarised and exhibited a lower rheobase, suggesting increased excitability and sensitivity to depolarising inputs. These changes may be linked to mechanical stress during transportation, which prompts microglia to release reactive oxygen species and cytokines (Banati et al., 1993). Microglia and astrocytes both have been shown to release pro-inflammatory cytokines in response to insults and neuronal inflammation, which was demonstrated to

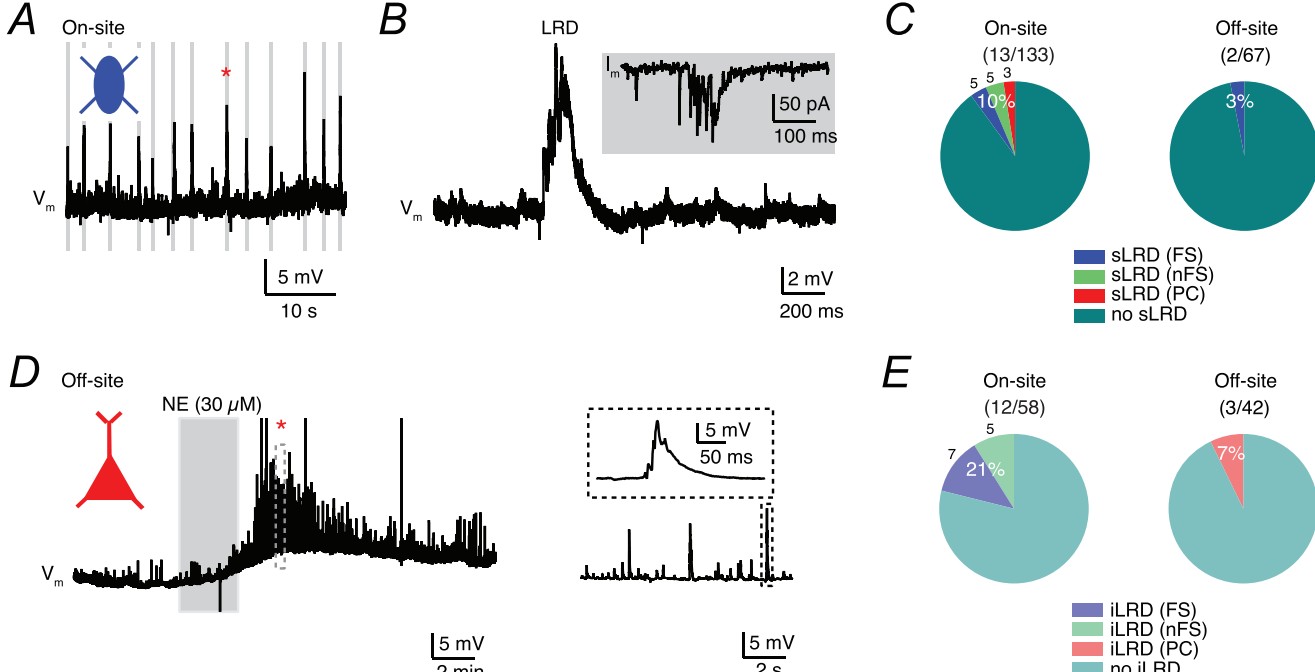

**Figure 7. Self-organized network activity was predominantly observed in human cortical neurons recorded on-site**
*A*, $V_m$ recordings from a human cortical L2/3 fast-spiking interneuron demonstrating spontaneously occurring large rhythmic depolarisations (sLRDs). This neuron was recorded on-site at the Aachen laboratory. *B*, a single LRD marked by an asterisk in (*A*) is enlarged, which is composed of several excitatory postsynaptic potentials (EPSPs) at the rising phase. Inset shows voltage-clamp recording of a single LRD from the same neuron. *C*, percentage of on- and off-site recorded neurons displaying sLRDs. $P = 0.043$ for $\chi^2$ test without Yates correction and therefore significant. *D*, LRDs were induced by bath application of noradrenaline (10 or 30 μM) (iLRDs) in a human cortical L2/3 pyramidal cell that showed no sLRDs in normal aCSF. The region marked by an asterisk is enlarged on the right. Insert, zoom-in view of an individual LRD. *E*, percentage of neurons exhibiting iLRDs recorded on- and off-sites. $P = 0.031$ for $\chi^2$ test without Yates correction and therefore significant.

enhance neuronal excitability by increasing sodium channel currents (Delbridge et al., 2020; Guo et al., 2007; Vezzani & Viviani, 2015; Wang et al., 2007). In addition, in acute slices, microglia undergo ATP-dependent morphological changes (Berki et al., 2024). However, the time course of inflammatory cascades (hours to days) suggests that such mechanisms are unlikely to explain immediate transport-related changes observed in acute recordings but may contribute to deterioration during extended post-transport incubation. Future studies measuring inflammatory markers, together with electrophysiological parameters, would be needed to establish whether and when inflammatory processes contribute to transport-induced functional changes in human cortical tissue.

For FS interneurons, although there was no change in the $V_{rest}$, off-site recordings showed a narrower AP half-width and an increased amplitude compared to on-site recordings. This might result from the washout of residual neuromodulators in the intercellular space during transport, as neuromodulators such as norepinephrine, acetylcholine and dopamine

can modulate AP amplitude and width (Gorelova et al., 2002; Kawaguchi & Shindou, 1998; Qi & Feldmeyer, 2022). These alterations could influence overall network dynamics and cortical information processing.

## Impact of transport on synaptic function and network events

The reduced network activity after transportation could have important implications for the study of human brain tissue, where intact network dynamics are crucial for understanding the Physiology and Pathophysiology of underlying mechanisms (de la Prida & Huberfeld, 2019; Nimmrich et al., 2015). Our analysis of sEPSPs indicates that transport-related effects on synaptic function are cell type-dependent. Although the frequency of sEPSPs did not differ significantly between on- and off-site recordings for both pyramidal cells and interneurons, there was a notable reduction in the mean EPSP amplitude in off-site recorded interneurons due to the reduction in large-amplitude EPSPs. This suggests that synaptic

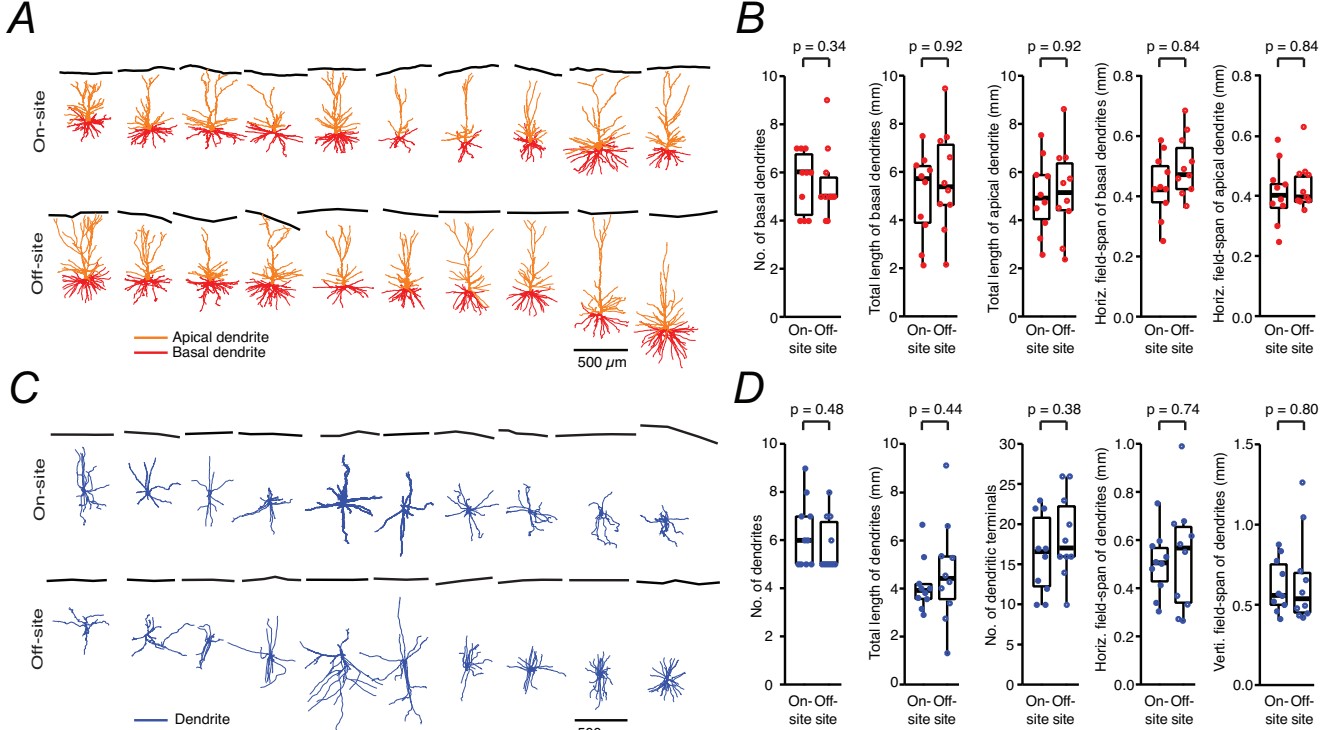

**Figure 8. Comparison of dendritic morphologies between on- and off-site recorded human cortical neurons**
*A*, morphological reconstructions of human cortical L2/3 pyramidal cells recorded on-site (top, *n* = 10 from 5 surgeries) and off-site (bottom, *n* = 10 from 8 surgeries). *B*, comparison of morphological properties of pyramidal cells recorded on- and off-site. (*C*, *D*) same as *A* and *B* but for human cortical L2/3 FS interneurons recorded on-site (top, *n* = 10 from 9 surgeries) and off-site (bottom, *n* = 10 from 8 surgeries). Here, dendrites were almost completely reconstructed for both pyramidal cells and fast-spiking interneurons. The somatodendritic morphology of on- and off-site recorded neurons showed no significant difference. *P*-values were calculated using the Wilcoxon–Mann–Whitney two-sample rank test.

**Table 4. Comparison of morphological properties between on- and off-site recorded human cortical neurons**

| | L2/3 pyramidal cells | | | L2/3 fast-spiking interneurons | | |
|---|---|---|---|---|---|---|
| | On-site (*n* = 10) | Off-site (*n* = 10) | *P*-value | On-site (*n* = 10) | Off-site (*n* = 10) | *P*-value |
| No. of basal dendrites | 5.7 ± 1.7 | 4.9 ± 0.7 | 0.34 | 6.3 ± 1.4 | 5.8 ± 1.1 | 0.48 |
| No. of basal dendrite branches | 27.8 ± 9.5 | 31.3 ± 9.1 | 0.50 | 16.2 ± 4.9 | 18.5 ± 5.2 | 0.38 |
| Total length of basal dendrites (mm) | 4.9 ± 1.8 | 5.2 ± 1.6 | 0.92 | 4.1 ± 1.1 | 4.7 ± 2.2 | 0.44 |
| Horizontal field span of basal dendrites (mm) | 0.45 ± 0.14 | 0.47 ± 0.07 | 0.84 | 0.50 ± 0.13 | 0.55 ± 0.28 | 0.74 |
| Vertical field span of basal dendrites (mm) | 0.29 ± 0.06 | 0.35 ± 0.11 | 0.29 | 0.62 ± 0.16 | 0.65 ± 0.29 | 0.80 |
| No. of apical dendrite branches | 22.6 ± 7.8 | 24.9 ± 9.1 | 0.89 | NA | NA | NA |
| Total length of apical dendrite (mm) | 5.1 ± 1.7 | 5.2 ± 1.8 | 0.92 | NA | NA | NA |
| Horizontal field span of apical dendrite (mm) | 0.42 ± 0.12 | 0.41 ± 0.04 | 0.84 | NA | NA | NA |

*Note*: The *P*-value was calculated using the non-parametric Wilcoxon–Mann–Whitney two-sample rank test. Data are presented as mean ± SD.
Abbreviation: NA, not applicable.

efficacy, particularly in excitatory-to-inhibitory circuits, may be compromised during transport. The decreased EPSP amplitude could reflect reduced presynaptic release probability or postsynaptic receptor sensitivity. Recent evidence also suggests that human pyramidal neurons may exhibit enhanced NMDA receptor contributions to synaptic transmission compared to rodent neurons (Eyal et al., 2018; Hunt et al., 2023), potentially rendering these synapses more vulnerable to metabolic or ionic perturbations during transport, though this remains to be directly tested.

Furthermore, one of the most noteworthy findings of our study is the significant reduction in the occurrence of sLRDs and NE-induced LRDs in neurons recorded off-site compared to those recorded on-site. Specifically, only 3% of neurons recorded off-site exhibited sLRDs compared to 10% on-site. Similarly, the percentage of neurons showing NE-induced LRDs was significantly lower off-site than on-site (21%). As LRDs are predominantly observed in human interneurons rather than in pyramidal cells and depend on glutamatergic transmission (Yang et al., 2024), the decreased EPSP amplitude found in off-site

recorded interneurons may serve as an indicator of reduced LRD occurrence. The reduction in pyramidal cell spine densities could be another contributing factor. This is supported by the fact that transport to an off-site laboratory has an adverse impact on network-level events, possibly due to subtle changes in the synaptic connectivity and neuromodulatory environment of neurons. It can be hypothesised that vibrations occurring during transport could therefore diminish the concentrations of neuro-modulators, such as NE, in the intercellular space. This could, in turn, substantially affect the occurrence of network events such as LRDs (Yang et al., 2024). To test this in principle, we used exogenous bath application of NE (30 µM) to induce LRD in neurons that did not display LRDs in normal aCSF, as reported previously (Yang et al., 2024). To further investigate whether altered neuromodulator signalling contributes to the reduction in LRD observed in off-site recorded neurons, a small set of experiments was performed by perfusing slices with human CSF. The preliminary data showed that the human CSF was able to induce LRD-like synaptic activity in human cortical neurons that did not exhibit LRDs in aCSF.

### Potential mechanisms underlying transport-induced changes

Transport-related mechanical stress likely affects neuronal properties through multiple mechanisms acting on different time scales and can have various detrimental effects on human brain tissue, largely due to its impact on the mechanical properties of the tissue. During transport, brain slices are continuously exposed to vibrations, acceleration, deceleration and sudden jolts, which can exert mechanical stress on the delicate structures. These forces can alter the mechanical properties of the brain tissue, which are determined not only by cellular components but also by the extracellular matrix (ECM). The ECM, comprising elements like collagen (Budday et al., 2020; Shulyakov et al., 2011), proteoglycans (Lotz & Loeser, 2012) and lipids (Mihai et al., 2017), provides

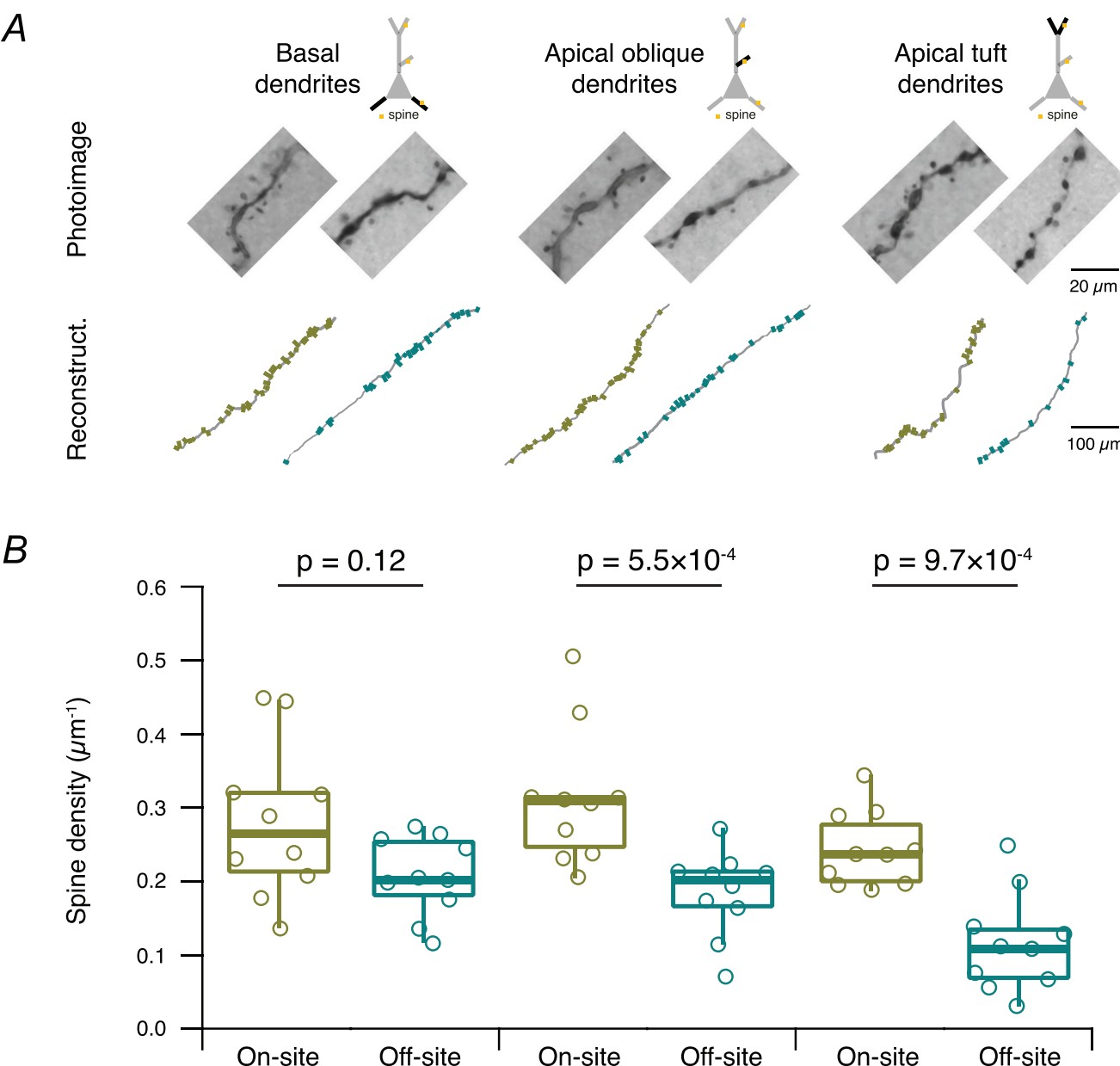

**Figure 9. Comparison of dendritic spine densities between on- and off-site recorded human cortical L2/3 pyramidal cells**

*A*, representative images and reconstructions of spines at basal, apical oblique and apical tuft dendrites for on- and off-site recorded L2/3 pyramidal cells after biocytin staining. *B*, comparison of spine densities at the three dendritic compartments for on-site ($n$ = 10 from 6 surgeries) and off-site ($n$ = 10 from 7 surgeries) recorded L2/3 pyramidal cells. Significant reductions in spine densities were found at the apical oblique and tuft dendrites for off-site recorded L2/3 pyramidal cells. *P*-values were calculated using the Wilcoxon–Mann–Whitney two-sample rank test.

crucial structural and biochemical support, influencing tissue stiffness, elasticity and resilience. The mechanical stress encountered during transport could alter these ECM components, leading to changes in tissue hydration, viscoelastic properties and membrane fluidity, ultimately affecting cellular behaviours such as signalling pathways.

Mechanical trauma causes immediate, depth-dependent neuronal damage resulting in increased plasma membrane permeability, calcium overload and calpain-mediated cell death occurring within minutes to hours (Chen & Lesnefsky, 2015; Geddes et al., 2003; Suryavanshi et al., 2024). The surge in calcium can additionally be triggered by reactive oxygen and nitrogen species (ROS/RNS) and could induce abnormal homeostatic responses, further altering neuronal properties (Higgins et al., 2010).

A second critical process is ATP depletion. ATP biosensors show that ATP is released after slicing and declines rapidly in the first 10 min (Berki et al., 2024). In addition, insufficient oxygenation during transport may enhance metabolic stress by creating hypoxic regions (Andersson et al., 1998). This leads to ionic dysregulation, with extracellular potassium concentrations increasing more than 10-fold within 3–10 min of metabolic stress (Hansen, 1988). Furthermore, even mild mechanical stress can dramatically elevate the production of ROS/RNS (Arundine et al., 2004). Additionally, mechanical stress can lead to a reduction in dendritic spines due to NMDA receptor-mediated increases in postsynaptic calcium (Chen & Lesnefsky, 2015). The decrease in spine density, and consequently synapse number, particularly near the apical dendrites, may contribute further to disrupted network activity. Oxidative stress can also disrupt the cytoskeletal elements, especially actin filaments (Allani et al., 2004), which play a critical role in the formation and regulation of dendritic spines (Lei et al., 2016).

## Implications for experimental design, future research and conclusion

The findings of this study highlight how transport-induced stress can lead to significant alterations in synaptic efficacy and network dynamics, notably the reduction in EPSP amplitudes of FS interneurons and LRDs. These changes, which may be induced by mechanical stress and its impact on the ECM and neuromodulator availability, can compromise the reliability of experimental data, especially in studies focused on synaptic plasticity or network events. On-site recordings, where transport-induced effects are minimised, are likely to yield more accurate and physiologically relevant results. It should be noted that, in addition to transport itself, variability in slice preparation, transport, experimental set-up, experimenter and the time after slicing may also influence the final results.

To mitigate these effects, future research should aim to refine transport protocols. Enhancing transport conditions – such as adjusting temperature, oxygenation and medium composition – could preserve tissue integrity. For example, transporting tissue blocks instead of slices may reduce mechanical stress on individual cells and help retain neuromodulators. However, hypoxia during transport remains a challenge that may be alleviated through hypothermia, which has been shown to preserve tissue better (Antonic et al., 2014). Because microglia are well known to be activated by physical stress and infection (Ayata & Schaefer, 2020; Quan & Zhang, 2023), we hypothesised that many of the observed effects could result from microglia-mediated changes, such as their established roles in synaptic pruning (Gunner et al., 2019; Paolicelli et al., 2011; Schafer et al., 2012). Therefore, potential improvements – such as using an anti-vibration box to store the slice holder during transport to reduce mechanical stress, and maintaining sterile conditions during slice preparation, transport and storage – would likely be beneficial.

In conclusion, although the firing capacity of human cortical neurons remains largely intact after transport, significant changes in neuronal excitability, synaptic efficacy and network-level events underscore the importance of considering transport effects in experimental designs. By optimising transportation protocols and exploring the molecular changes induced by mechanical stress, researchers can better preserve tissue integrity. This will ultimately improve the quality and reliability of human brain tissue research, advancing our understanding of cortical microcircuits and their role in both healthy and diseased states.

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

## Additional information

### Data availability statement

The data that support the findings of this study are available from the corresponding author upon reasonable request.

### Competing interests

All authors confirm that there are no relevant financial or non-financial competing interests to report.

### Author contributions

G.Q.: investigation, formal analysis, data curation, visualization, validation, methodology, writing – review & editing, conceptualization. D.Y.: investigation, formal analysis, data curation, visualization, validation, methodology, writing – review & editing, conceptualization. B.A.V.: writing – review & editing, writing – original draft, visualization, validation, methodology, investigation. H.W.: validation, methodology. H.H.: writing – review & editing, validation. D.D.: writing – review & editing, validation. F.D.: writing – review & editing, writing – original draft, validation, supervision, resources, project administration, methodology, funding acquisition, conceptualization. K.H.: writing – review & editing, writing – original draft, validation, supervision, resources, project administration, methodology, funding acquisition, conceptualization.

### Funding

This work was supported by funding from the European Union's Horizon 2020 Framework Program for Research and Innovation under the Framework Partnership Agreement (HBP FPA) no. 650003 and by the Chan Zuckerberg Initiative Collaborative Pairs Pilot Project Awards (Phase 1 and Phase 2).

### Acknowledgements

The authors would like to thank Dr. Karlijn van Aerde for custom-written macros using Igor Pro software for electrophysiological analysis, and Xuangui Zhang and Yutong Wu for their assistance in neuronal morphological reconstructions.

### Keywords

human cortical neurons, intrinsic firing properties, synaptic properties

## Supporting information

Additional supporting information can be found online in the Supporting Information section at the end of the HTML view of the article. Supporting information files available:

**Peer Review History**

