## [Peer Review History · The Journal of Physiology]

Transport-Related Effects on Intrinsic and Synaptic Properties of Human Cortical Neurons: A Comparative Study

Guanxiao Qi, Danqing Yang, Aniella Vanessa Bak, Werner Hucko, Daniel Delev, Hussam Aldin Hamou, Dirk Feldmeyer, and Henner Koch

DOI: 10.1113/JP288111

Corresponding author(s): Henner Koch (hkoch@ukaachen.de)

Review Timeline:

Submission Date:	15-Nov-2024
Editorial Decision:	07-Feb-2025
Revision Received:	06-Nov-2025
Editorial Decision:	25-Nov-2025
Revision Received:	05-Dec-2025
Accepted:	08-Dec-2025

Senior Editor: Katalin Toth

Reviewing Editor: Katalin Toth

Transaction Report:

Dear Dr Koch,

Re: JP-RP-2024-288111 "Transport-Related Effects on Intrinsic and Synaptic Properties of Human Cortical Neurons: A Comparative Study" by Guanxiao Qi, Danqing Yang, Aniella Bak, Werner Hucko, Daniel Delev, Hussam Aldin Hamou, Dirk Feldmeyer, and Henner Koch

Thank you for submitting your manuscript to The Journal of Physiology. It has been assessed by a Reviewing Editor and by 2 expert referees and we are pleased to tell you that it is potentially acceptable for publication following satisfactory major revision.

LANGUAGE EDITING AND SUPPORT FOR PUBLICATION: If you would like help with English language editing, or other article preparation support, Wiley Editing Services offers expert help, including English Language Editing, as well as translation, manuscript formatting, and figure formatting at www.wileyauthors.com/eoo/preparation. You can also find resources for Preparing Your Article for general guidance about writing and preparing your manuscript at www.wileyauthors.com/eoo/prepresources.

REVISION CHECKLIST:

We look forward to receiving your revised submission.

Yours sincerely,

Katalin Toth
Senior Editor
The Journal of Physiology

REQUIRED ITEMS

- Include a Key Points list in the article itself, before the Abstract.
- Author photo and profile. First or joint first authors are asked to provide a short biography (no more than 100 words for one author or 150 words in total for joint first authors) and a portrait photograph. These should be uploaded and clearly labelled together in a Word document with the revised version of the manuscript. See Information for Authors for further details.
- You must start the Methods section with a paragraph headed Ethical Approval. If experiments were conducted on humans, confirmation that informed consent was obtained, preferably in writing, that the studies conformed to the standards set by the latest revision of the Declaration of Helsinki and that the procedures were approved by a properly constituted ethics committee, which should be named, must be included in the article file. If the research study was registered (clause 35 of the Declaration of Helsinki), the registration database should be indicated, otherwise the lack of registration should be noted as an exception (e.g. The study conformed to the standards set by the Declaration of Helsinki, except for registration in a database). For further information see: <https://physoc.onlinelibrary.wiley.com/hub/human-experiments>.
- The Journal of Physiology funds authors of provisionally accepted papers to use the premium BioRender site to create high resolution schematic figures. Follow this link and enter your details and the manuscript number to create and download figures. Upload these as the figure files for your revised submission. If you choose not to take up this offer, we require figures to be of similar quality and resolution. If you are opting out of this service to authors, state this in the Comments section on the Detailed Information page of the submission form. The link provided should only be used for the purposes of this submission. Authors will be charged for figures created on this premium BioRender account if they are not related to this manuscript submission.
- Please upload separate high-quality figure files via the submission form.

- Please ensure that the Article File you upload is a Word file.

- Your paper contains Supporting Information of a type that we no longer publish, including supplementary tables and figures. Any information essential to an understanding of the paper must be included as part of the main manuscript and figures. The only Supporting Information that we publish are video and audio, 3D structures, program codes and large data files. Your revised paper will be returned to you if it does not adhere to our Supporting Information Guidelines.

- Papers must comply with the Statistics Policy: https://jp.msubmit.net/cgi-bin/main.plex?form_type=display_requirements#statistics.

In summary:

- If $n \leq 30$, all data points must be plotted in the figure in a way that reveals their range and distribution. A bar graph with data points overlaid, a box and whisker plot or a violin plot (preferably with data points included) are acceptable formats.

- If $n > 30$, then the entire raw dataset must be made available either as supporting information, or hosted on a not-for-profit repository, e.g. FigShare, with access details provided in the manuscript.

- 'n' clearly defined (e.g. x cells from y slices in z animals) in the Methods. Authors should be mindful of pseudoreplication.

- All relevant 'n' values must be clearly stated in the main text, figures and tables.

- The most appropriate summary statistic (e.g. mean or median and standard deviation) must be used. Standard Error of the Mean (SEM) alone is not permitted.

- Exact p values must be stated. Authors must not use 'greater than' or 'less than'. Exact p values must be stated to three significant figures even when 'no statistical significance' is claimed.

Reviewing Editor's comments:

Dear authors, our apologies for the long review duration. We lost one reviewer in the review process and needed to find a new, third reviewer.

As the reviewers point out, single cell electrophysiology and morphology from human cortical tissue is very valuable. However, the low n-numbers and missing paired statistical evaluation do not yet allow sufficient support of your conclusions.

Referee #1:

Summary:

In this study, Qi et al., have compared the intrinsic, synaptic and morphological properties of human cortical neurons in acute brain slices under two conditions: with or without 30-40 min transportation in a custom-made slice keeper. They used patch-clamp recordings and morphological reconstruction of cortical neurons and describe specific changes in slices recorded off-site after transportation. Specifically, they found increased excitability of pyramidal neurons and action potential kinetics of fast-spiking interneurons. Spontaneous EPSP amplitudes in interneurons and overall network-driven events were reduced. Finally, they found a reduced spine density of pyramidal neurons.

Electrophysiological recordings from human tissue are a very important tool to understand fundamental principles of human neuronal and synaptic function. Thus, identifying and avoiding potential methodological biases introduced by specific steps of tissue preparation is of great importance and relevance to the field. While the overall approach and the different methods chosen in this study are valid and performed at high quality, the study would benefit from a more rigorous statistical analysis and may have overlooked additional confounding factors.

Major comments:

1. While the abstract mentions $n = 200$ neurons from 32 surgeries, most results only had 10 to 11 data points per group (except for Fig. 7). This is rather low, especially considering the (known) large variance of intrinsic and synaptic properties, which can be also seen in this study (e.g. Fig. 4B). Varying tissue quality across surgical samples and slices can greatly impact the results. Consequently, the level of independent replication should be on the level of surgeries/transportations. As data were recorded at both locations, the data visualization should take this into account and the statistics should be performed in a paired fashion. For example, the author could perform a linear mixed effect model and include surgery as a random factor and plot the different slopes for each surgery. This comment applies to all quantifications in the study and the approach would likely require additional experiments to reach statistical significance. In addition, the authors might consider where correction for multiple hypothesis testing would be warranted.
2. Another potential confounder that has not been mentioned or discussed are systematic biases introduced by using different setups or having different experimenters. More details in the recording steps would be helpful, for example clarify that the technical equipment is the same and to what extent all experimental steps, such as pipette pulling, neuron selection and patching, temperature during recordings, compensation of bridge balance and capacitance have been standardized. Especially, differences in capacitance compensation and series resistance (from different pipette shapes) could systematically affect fast signals, such as the action potential of fast-spiking interneurons. To address this, control experiments on both setups using slices with no difference in transport would be necessary. For example, rodent slices without transport or recording human tissue after transport also at the on-site laboratory.
3. The time of experiment after slicing has not been sufficiently addressed. While Fig. 2 and Fig. S2 show changes up to 3 days after resection, the implication of these results for the overall study and research question is not clear. Do the authors see the same trends/differences in the on-site tissue? More importantly, at what times were the different data points from the other figures recorded? Can the authors (statistically) exclude the possibility that the reported differences are not due to different experimental time points of on-site vs off-site recordings?
4. The publication that originally describes the large rhythmic depolarizations (LRD, Yang et al., Communications Biology 2024) reports different proportions of cells that are LRD+ in the pyramidal- versus the interneuron population. Within the interneuron population, there are further differences in the fraction of LRD+ cells between smaller and larger basket-cell-like cells. In the current study, the neuron type for the LRD+/LRD- neurons is not specified. The authors should thus discriminate the cell type in their analysis. Also, the authors state that "... neurons displaying spontaneous LRDs that were recorded at the off-site laboratory are located significantly deeper in the slice than those neurons showing no LRD". Are the recording depths the same on-site versus off-site?

Minor comments:

1. Claims in key points should be attenuated and adapted:
 - a. First key point: The use of the word "significantly" is not ideal, as it is typically used in the context of statistical significance of a specific comparison.
 - b. Second key point: The authors present no empirical evidence for a neuromodulator wash-out mediated effect (see comment below).
 - c. Fourth key point: It is not clear how the results of the study "advance our understanding of cortical microcircuitry" beyond the raised methodological effects.
2. Fig S2: Could the increase in EPSP amplitude and the decrease in EPSP frequency be interrelated and biased by potential differences in noise? I assume each data point is an average from one neuron, please clarify how many events or what time period was used for the averaging. Maybe the study could benefit from controlling for detection limits as shown in this preprint: ['Mini analysis' is an unreliable reporter of synaptic changes. Ingo H. Greger, Jake F. Watson. bioRxiv 2024.10.26.620084]
3. Fig4: How was the maximum firing frequency determined? What was the stimulation amplitude? The on-site group looks surprisingly homogeneous. Can you show an FI-slope? How do these values compare to previous reports, such as [Berg, J., Sorensen, S.A., Ting, J.T. et al. Human neocortical expansion involves glutamatergic neuron diversification. Nature 598, 151-158 (2021)] or [Planert et al., Cellular and Synaptic Diversity of Layer 2-3 Pyramidal Neurons in Human Individuals. bioRxiv 2021.11.08.467668] ?
4. Fig 6: Color legend and/or schematic would help to make clear that it is PC vs FS comparison.
5. Fig6: To what extent can changes on the synaptic level be explained by changes in neuronal excitability? The statement that differences are specific to interneurons is quite strong, considering potential confounds (slice variance) and low sample size. Were pyramidal neurons and interneurons from the same tissue compared?
6. P20: Here it says "off-site" twice: "... the amplitude of EPSPs in interneurons was significantly reduced when recordings are conducted off-site compared to off-site recordings."
7. Fig 7C typo ($p=p=$).
8. Fig7C and 7E: "n.s. $p > 0.05$ " is bit confusing, suggesting that the comparisons were not significant while they are. Maybe rephrase.
9. Fig7C reports 2/67 neurons for the off-site group. However, the text states $n=2$ for LRD+ and $n=30$ for LRD- neurons (page 21). This difference in "n's" should be resolved.
10. P21: Spontaneous LRD are found in "significantly" deeper neurons. "Significantly" is a strong word considering that only $n = 2$ of neurons with LRD+ were recorded. Is there a statistical confirmation? Was soma depth also predictive for NE-responsiveness?
11. Fig 8 and Table 3: A visualization of the quantification in the figure (bar plots with scatter) could help convey the message and be more consistent with other results.
12. Fig 9: Similar to general comment to other figures: From how many samples were the neurons taken? Does n represent region/dendrite imaged or individual neurons? From how many surgeries were these data collected?
13. P25: "comprehensive" is quite a strong word for $n = 10$ in each condition.
14. P26: If reduction in LRDs is "most noteworthy", consider expanding on this result a bit more. What about amplitude and frequency of LRD? If LRDs are predominantly reported in interneurons, how many of the data in Fig7C were from

interneurons and how many were PCs?

15. P26: Why is the reduction in LRD due to neuromodulator wash-out? Could the reduction not be well in line with the detected synaptic and spine changes? Overall, I do not see much support for the proposed neuromodulator wash-out hypothesis. It is likely that a few minutes of perfusion in the recording chamber would already substitute most of compounds in the extracellular solution, considering the kinetics of typical pharmacological wash-in and wash-out experiments. Furthermore, the result in Fig. 7 suggests rather the opposite: NE condition increased LRD in both conditions by roughly 200%. If the transport-related changes would be due to neuromodulator wash-out, I would expect that NE should have a differential and rescuing effect on the off-site slices.

16. If the authors have included data from later days and considering the non-sterile conditions, it is likely that bacterial growth has occurred that could have activated microglia. While microglia-activation has been briefly mentioned for the discussion of intrinsic properties, one could also hypothesize that most of the observed effects could arise from microglia mediated changes, for example due their established role in synaptic pruning. Because it is well established that microglia are activated by physical stress (and infection), the discussion on potential mechanisms and improvement should elaborate more on these aspects. The benefits of sterile preparation, transport and slice storage conditions as well as potential improvements to reduce mechanical stress during transport should be discussed.

Referee #2:

Single cell electrophysiology of human cortical neurons is rare and thus very valuable. However, at present, the data do not convincingly support the result, that differences measured on-site vs off-site are really due to the transport. As patch-clamp recordings usually follow a tightly controlled experimental design, a data set of $n=10$ or 11 should be sufficient if recorded at the same setup by the same experimenter. Since this is not the case in your study, you would 1) need a significantly larger data set, 2) indicate pairs of neurons that originate from the same probe (link the dots in the graphs by lines or color code the tissue samples), and 3) perform paired statistics. Comparing data from the same probe is of outmost importance since I suppose neurons from epilepsy patients maybe more excitable than those of tumor patients and could induce bias independent of transport. Once color/symbol coded, you could state in the figure legends from which sample the neurons were recorded.

I suggest the following experiment to control for changes due to transport (independent of confounding factors such as a second setup or another experimenter). You treat a batch of tissue samples exactly as if transporting them to the off-site location, but then drive around for the same time and return to the on-site location where the same setup and same experimenter will perform the "post-transport" experiments.

An important take-home message from your study also outside possible medical applications is the fact that you can keep brain slices alive and healthy for up to 60 hours and share them between reasonable locations. This is an important point to make as it will be a revolution regarding reducing animal numbers in research (3Rs) and scientist could share rodent brain slices across campus or even further.

I would also like to add a few minor comments.

In your key points, rather than stating "changes of neuronal excitability", please state whether neuronal excitability increase or decreases.

The change in neuromodulator action has not been tested in your experiments, but is a discussion point. Though a valid discussion point, it is not one of your key findings. Narrower action potentials could for example also be caused by slight differences in the recording temperature between the two setups.

Key points 3 and 4 are again conclusions rather than results of your study. I agree it is an important point to make, but these two points should be combined so that out of the 4 key points are at least three that state your findings.

Your rationale of using box plots for $n > 10$ and bar graphs for $n < 10$ is uncommon and not consistent in your figures (for example fig 6B should be bar graphs according to your rule). Usually bar graphs are used when data are not normally distributed, while box plots are suited for normally distributed data. As you used non-parametric statistics, I recommend to use box plots in all your figures, instead of bar graphs.

In your figure 5B, you report the off-site APs to be briefer but larger in amplitude. Independent of where or when APs were recorded, smaller AP half-width usually is due to stronger contribution of high-voltage gated potassium currents, but if that is the case, I would expect the AP amplitudes to be the same or smaller rather than larger. Perhaps in your case the change in half-width is due to a change in AP rise time rather than repolarization. You could provide these measures for the data you have and thereby hint at possible mechanisms.

In general, rather than making a strong point about the transport, maybe highlight, the variability that can occur due to different factors such as transport, setups, experimenter, time post tissue harvest etc.

END OF COMMENTS

Summary

In this study, Qi et al., have compared the intrinsic, synaptic and morphological properties of human cortical neurons in acute brain slices under two conditions: with or without 30-40 min transportation in a custom-made slice keeper. They used patch-clamp recordings and morphological reconstruction of cortical neurons and describe specific changes in slices recorded off-site after transportation. Specifically, they found increased excitability of pyramidal neurons and action potential kinetics of fast-spiking interneurons. Spontaneous EPSP amplitudes in interneurons and overall network-driven events were reduced. Finally, they found a reduced spine density of pyramidal neurons.

Electrophysiological recordings from human tissue are a very important tool to understand fundamental principles of human neuronal and synaptic function. Thus, identifying and avoiding potential methodological biases introduced by specific steps of tissue preparation is of great importance and relevance to the field. While the overall approach and the different methods chosen in this study are valid and performed at high quality, the study would benefit from a more rigorous statistical analysis and may have overlooked additional confounding factors.

Major comments:

1. While the abstract mentions $n = 200$ neurons from 32 surgeries, most results only had 10 to 11 data points per group (except for Fig. 7). This is rather low, especially considering the (known) large variance of intrinsic and synaptic properties, which can be also seen in this study (e.g. Fig. 4B). Varying tissue quality across surgical samples and slices can greatly impact the results. Consequently, the level of independent replication should be on the level of surgeries/transports. As data were recorded at both locations, the data visualization should take this into account and the statistics should be performed in a paired fashion. For example, the author could perform a linear mixed effect model and include surgery as a random factor and plot the different slopes for each surgery. This comment applies to all quantifications in the study and the approach would likely require additional experiments to reach statistical significance. In addition, the authors might consider where correction for multiple hypothesis testing would be warranted.
2. Another potential confounder that has not been mentioned or discussed are systematic biases introduced by using different setups or having different experimenters. More details in the recording steps would be helpful, for example clarify that the technical equipment is the same and to what extent all experimental steps, such as pipette pulling, neuron selection and patching, temperature during recordings, compensation of bridge balance and capacitance have been standardized. Especially, differences in capacitance compensation

and series resistance (from different pipette shapes) could systematically affect fast signals, such as the action potential of fast-spiking interneurons. To address this, control experiments on both setups using slices with no difference in transport would be necessary. For example, rodent slices without transport or recording human tissue after transport also at the on-site laboratory.

3. The time of experiment after slicing has not been sufficiently addressed. While Fig. 2 and Fig. S2 show changes up to 3 days after resection, the implication of these results for the overall study and research question is not clear. Do the authors see the same trends/differences in the on-site tissue? More importantly, at what times were the different data points from the other figures recorded? Can the authors (statistically) exclude the possibility that the reported differences are not due to different experimental time points of on-site vs off-site recordings?
4. The publication that originally describes the large rhythmic depolarizations (LRD, Yang et al., Communications Biology 2024) reports different proportions of cells that are LRD+ in the pyramidal- versus the interneuron population. Within the interneuron population, there are further differences in the fraction of LRD+ cells between smaller and larger basket-cell-like cells. In the current study, the neuron type for the LRD+/LRD- neurons is not specified. The authors should thus discriminate the cell type in their analysis. Also, the authors state that "... neurons displaying spontaneous LRDs that were recorded at the off-site laboratory are located significantly deeper in the slice than those neurons showing no LRD". Are the recording depths the same on-site versus off-site?

Comment [MF1]: Habe mir das originale Paper mal angekuckt. Das scheint mir ein nicht irrelevanter Punkt zu sein. Aber meinerwegen kann man das auch zu einem "minor comment" machen.

Minor comments:

1. Claims in key points should be attenuated and adapted:
 - a. First key point: The use of the word "significantly" is not ideal, as it is typically used in the context of statistical significance of a specific comparison.
 - b. Second key point: The authors present no empirical evidence for a neuromodulator wash-out mediated effect (see comment below).
 - c. Fourth key point: It is not clear how the results of the study "advance our understanding of cortical microcircuitry" beyond the raised methodological effects.
2. Fig S2: Could the increase in EPSP amplitude and the decrease in EPSP frequency be interrelated and biased by potential differences in noise? I assume each data point is an average from one neuron, please clarify how many events or what time period was used for the averaging. Maybe the study could benefit from controlling for detection limits as shown in this preprint: [‘Mini analysis’ is an unreliable reporter of synaptic changes. Ingo H. Greger, Jake F. Watson. bioRxiv 2024.10.26.620084]

3. Fig4: How was the maximum firing frequency determined? What was the stimulation amplitude? The on-site group looks surprisingly homogeneous. Can you show an FI-slope? How do these values compare to previous reports, such as [Berg, J., Sorensen, S.A., Ting, J.T. *et al.* Human neocortical expansion involves glutamatergic neuron diversification. *Nature* **598**, 151–158 (2021)] or [Planert et al., Cellular and Synaptic Diversity of Layer 2-3 Pyramidal Neurons in Human Individuals. bioRxiv 2021.11.08.467668] ?
4. Fig 6: Color legend and/or schematic would help to make clear that it is PC vs FS comparison.
5. Fig6: To what extent can changes on the synaptic level be explained by changes in neuronal excitability? The statement that differences are specific to interneurons is quite strong, considering potential confounds (slice variance) and low sample size. Were pyramidal neurons and interneurons from the same tissue compared?
6. P20: Here it says “off-site” twice: “... the amplitude of EPSPs in interneurons was significantly reduced when recordings are conducted off-site compared to off-site recordings.”
7. Fig 7C typo ($p=p=$).
8. Fig7C and 7E: “n.s. $p > 0.05$ ” is bit confusing, suggesting that the comparisons were not significant while they are. Maybe rephrase.
9. Fig7C reports 2/67 neurons for the off-site group. However, the text states $n=2$ for LRD+ and $n=30$ for LRD- neurons (page 21). This difference in “n’s” should be resolved.
10. P21: Spontaneous LRD are found in “significantly” deeper neurons. “Significantly” is a strong word considering that only $n = 2$ of neurons with LRD+ were recorded. Is there a statistical confirmation? Was soma depth also predictive for NE-responsiveness?
11. Fig 8 and Table 3: A visualization of the quantification in the figure (bar plots with scatter) could help convey the message and be more consistent with other results.
12. Fig 9: Similar to general comment to other figures: From how many samples were the neurons taken? Does n represent region/dendrite imaged or individual neurons? From how many surgeries were these data collected?
13. P25: “comprehensive” is quite a strong word for $n = 10$ in each condition.
14. P26: If reduction in LRDs is “most noteworthy”, consider expanding on this result a bit more. What about amplitude and frequency of LRD? If LRDs are predominantly reported in interneurons, how many of the data in Fig7C were from interneurons and how many were PCs?

Comment [MF2]: Hab nur die Reihenfolge umgedreht, damit wir nicht mit uns selbst anfangen.

15. P26: Why is the reduction in LRD due to neuromodulator wash-out? Could the reduction not be well in line with the detected synaptic and spine changes? Overall, I do not see much support for the proposed neuromodulator wash-out hypothesis. It is likely that a few minutes of perfusion in the recording chamber would already substitute most of compounds in the extracellular solution, considering the kinetics of typical pharmacological wash-in and wash-out experiments. Furthermore, the result in Fig. 7 suggests rather the opposite: NE condition increased LRD in both conditions by roughly 200%. If the transport-related changes would be due to neuromodulator wash-out, I would expect that NE should have a differential and rescuing effect on the off-site slices.
16. If the authors have included data from later days and considering the non-sterile conditions, it is likely that bacterial growth has occurred that could have activated microglia. While microglia-activation has been briefly mentioned for the discussion of intrinsic properties, one could also hypothesize that most of the observed effects could arise from microglia mediated changes, for example due their established role in synaptic pruning. Because it is well established that microglia are activated by physical stress (and infection), the discussion on potential mechanisms and improvement should elaborate more on these aspects. The benefits of sterile preparation, transport and slice storage conditions as well as potential improvements to reduce mechanical stress during transport should be discussed.

NOTE: The comments from Editor and Reviewers are in black and the responses from Authors are in blue.

Reviewing Editor's comments:

Dear authors, our apologies for the long review duration. We lost one reviewer in the review process and needed to find a new, third reviewer.

As the reviewers point out, single cell electrophysiology and morphology from human cortical tissue is very valuable. However, the low n-numbers and missing paired statistical evaluation do not yet allow sufficient support of your conclusions.

Response to the Editor's Comments: We thank the editor for the positive feedback on the paper. In the revised manuscript, the sample sizes (n-numbers) have been substantially increased. For example, in the electrophysiology analysis of human L2/3 pyramidal cells in Fig. 4, the number of neurons in on-site recordings was increased from n = 10 to 25, and for off-site recordings from n = 11 to 20. Additionally, in Figs. 5, 6, 8, and 9, the number of recorded neurons has been increased to 10 or more. Interestingly, despite the increase in our sample size, the main conclusions remain unchanged. For specific data, such as the electrophysiology of human L2/3 pyramidal cells, paired statistical tests were performed (see marked sections in the manuscript and Fig. R1).

In addition to the specific point-to-point reply to the reviewers below, we would like to summarize the main changes to the manuscript here:

- (1) A list of Key Points has been added to the article before the Abstract.
- (2) The Author's photo and profile have been prepared.
- (3) A paragraph headed "Ethical Approval" has been added to the Methods section, including the required information for human-related experiments.
- (4) Separate high-quality figure files, along with the Article File in Word format, have been uploaded via the submission forum.
- (5) The Supporting Information from the previous version of the manuscript has either been incorporated into the main figure or removed entirely.
- (6) All the figures are now presented as box plots (showing box, whiskers, and median value) with individual data points overlaid. All relevant 'n' values are now clearly stated in the main text, figures, and tables.
- (7) Exact p-values have been provided throughout the main text, figures, and tables, even in cases where 'no statistical significance' is claimed.

Reviewer #1

Summary:

In this study, Qi et al., have compared the intrinsic, synaptic and morphological properties of human cortical neurons in acute brain slices under two conditions: with or without 30-40 min transportation in a custom-made slice keeper. They used patch-clamp recordings and morphological reconstruction of cortical neurons and describe specific changes in slices recorded off-site after transportation. Specifically, they found increased excitability of pyramidal neurons and action potential kinetics of fast-spiking interneurons. Spontaneous EPSP amplitudes in interneurons and overall network-driven events were reduced. Finally, they found a reduced spine density of pyramidal neurons.

Electrophysiological recordings from human tissue are a very important tool to understand fundamental principles of human neuronal and synaptic function. Thus, identifying and avoiding potential methodological biases introduced by specific steps of tissue preparation is of great importance and relevance to the field. While the overall approach and the different methods chosen in this study are valid and performed at high quality, the study would benefit from a more rigorous statistical analysis and may have overlooked additional confounding factors.

Response: We sincerely appreciate your positive feedback on our paper. In line with your comments and suggestions, we have revised the manuscript as follows:

Reviewer #1

Major comments:

1. While the abstract mentions $n = 200$ neurons from 32 surgeries, most results only had 10 to 11 data points per group (except for Fig. 7). This is rather low, especially considering the (known) large variance of intrinsic and synaptic properties, which can be also seen in this study (e.g. Fig. 4B). Varying tissue quality across surgical samples and slices can greatly impact the results. Consequently, the level of independent replication should be on the level of surgeries/transports. As data were recorded at both locations, the data visualization should take this into account and the statistics should be performed in a paired fashion

Response to the reviewer:

We agree with the comment of the reviewer. Among the 200 neurons obtained from 32 surgeries, not all yielded high-quality electrophysiological or morphological data. Only recordings that passed our quality-control criteria were included in the subsequent analyses. For electrophysiology, the resting membrane potential had to be more negative than -60 mV, and the series resistance had to be less than 40 M Ω . In addition, comparative analyses were restricted to pyramidal cells and fast-spiking interneurons; non-fast-spiking interneurons were excluded. These requirements resulted in a relatively limited sample size for each comparison. In the revised manuscript, the number of experiments has been substantially increased by re-analysing existing patch-clamp data by loosening the restriction for the series resistance (R_s) from previously ≤ 40 M Ω to now ≤ 50 M Ω , performing new

electrophysiological recordings, and adding further morphological reconstructions. For example, in the electrophysiological analysis of human L2/3 pyramidal cells (Fig. 4), the number of recorded neurons (n) was increased from n = 10 to 25 for on-site recordings, and from n = 11 to 20 for off-site recordings. In addition, for Figs. 5, 6, 8, 9, n was increased to 10 or more.

Reviewer #1: For example, the author could perform a linear mixed effect model and include surgery as a random factor and plot the different slopes for each surgery. This comment applies to all quantifications in the study and the approach would likely require additional experiments to reach statistical significance. In addition, the authors might consider where correction for multiple hypothesis testing would be warranted.

Response:

We agree that further statistical analysis improved the interpretation of the results of our study, and a linear mixed effects (LME) model was performed to analyse neuronal intrinsic properties of L2/3 pyramidal cells. We used the location of the recordings (on-site versus off-site) as the fixed effect and the surgery as a random effect to account for potential clustering within surgical sessions. The analysis revealed that 5 out of 12 parameters showed statistically significant differences between on-site and off-site: V_{rest} ($p = 1.4 \times 10^{-5}$), membrane time constant ($p = 7.5 \times 10^{-4}$), rheobase current ($p = 0.004$), AP amplitude ($p = 0.01$), and maximum firing frequency ($p = 1.5 \times 10^{-6}$). While other parameters, such as the input resistance R_{in} ($p = 0.53$), sag ($p = 0.14$), AP threshold ($p = 0.60$), AP rise time ($p = 0.30$) AP half-width ($p = 0.22$), AHP amplitude ($p = 0.092$), F-I slope ($p = 0.093$), showed no significant differences.

Fig. R1. Paired comparison of electrophysiological properties in human cortical L2/3 pyramidal cells recorded on- and off-site. Data points were calculated by averaging the parameters extracted from individual PCs obtained from the same surgery. In total, parallel on- and off-site recordings were obtained from six surgeries, yielding 20 PCs recorded on-site and 16 PCs recorded off-site. P values were calculated using the Wilcoxon signed-rank test for paired data.

Reviewer #1: 2. Another potential confounder that has not been mentioned or discussed are systematic biases introduced by using different setups or having different experimenters. More details in the recording steps would be helpful, for example clarify that the technical equipment is the same and to what extent all experimental steps, such as pipette pulling, neuron selection and patching, temperature during recordings, compensation of bridge balance and capacitance have been standardized. Especially, differences in capacitance compensation and series resistance (from different pipette shapes) could systematically affect fast signals, such as the action potential of fast-spiking interneurons. To address this, control experiments on both setups using slices with no difference in transport would be necessary. For example, rodent slices without transport or recording human tissue after transport also at the on-site laboratory.

Response: We agree with this potential systematic bias introduced by using different setups or having different experimenters. To address this concern, we added a new Table 2, which details the experimental setups and patch-clamp recording conditions at both sites. As shown in the table, only minor differences exist between the two setups. Furthermore, the experimenters – DY (on-site) and GQ (off-site) – were both trained in the same laboratory under the supervision of the same senior researcher (DF). Under these conditions, a systematic bias attributable to experimenters is unlikely. We added a section to the Methods and Results to describe this in detail:

Methods, Page 6: *“The setups used for on- and off-site recordings were nearly identical, with only minor differences, such as the light microscopy used for imaging (Table 2). Furthermore, the experimenters for on-site (DY) and off-site (GQ) recordings were trained in the same laboratory using the same patch-clamp setup. These efforts were taken to reduce bias related to the setup or the experimenter in the recording procedures.”*

Results Page 12: *“An objective comparison of on-site and off-site data may be influenced by systematic bias introduced using different setups or different experimenters. To minimize the potential for such bias, we used experimental setups at both sites that were very similar in composition (see Table 2). Additionally, the experimenters who conducted patch-clamp recordings at both on-site and off-site locations had very similar technical expertise. To further minimize bias related to setups and experimenters, all experiments were performed by the same experimenter using human cortical slices obtained from three surgeries. These experiments were conducted either directly in the on-site laboratory or after a 30–40 minute round-trip transport.”*

Because of local restrictions, it was not possible to perform rodent slice experiments with the on-site setup. Following the reviewer’s suggestion, however, additional experiments were carried out by the same experimenter on human cortical slices from three surgeries, processed either immediately or after a short round-trip transport (30–40 min) to the on-site laboratory. As summarized in Fig. R2 and Table R1, transported slices showed statistically significant changes in V_{rest} , action-potential (AP) half-width, and AP amplitude, mirroring the differences reported in Fig. 4B. By contrast, neither sEPSP frequency nor sEPSP amplitude differed significantly between transported and non-transported slices, in agreement with the results presented in Fig. 6B.

Fig. R2. Comparison of the intrinsic and synaptic properties of human cortical L2/3 pyramidal cells recorded in control and transported slices at the same setup (Aachen) by the same experimenter (GQ).

(A) Comparison of electrophysiological properties between neurons recorded in control ($n = 7$) and transported ($n = 7$) slices. (B) Comparison of spontaneous EPSP properties between neurons recorded in control ($n = 7$) and transported ($n = 7$) slices. P-values were calculated using the Wilcoxon-Mann-Whitney two-sample rank test.

Table R1. Comparison of electrophysiological properties between human cortical L2/3 pyramidal neurons recorded in control and transported slices at on-site setup (Aachen lab). P-values were calculated using the Wilcoxon signed-rank test.

	L2-3 pyramidal cell		
	Control ($n=7$)	Transported ($n=7$)	P value
Passive			
V_{rest} (mV)	-78.6 ± 3.5	-74.3 ± 2.0	0.026
R_{in} (MΩ)	101.9 ± 40.9	151.6 ± 63.4	0.13
Tau (ms)	18.8 ± 3.0	23.9 ± 8.8	0.21
Sag (%)	5.2 ± 4.5	5.3 ± 5.3	0.80
Single AP			
Rheobase current (pA)	238.6 ± 108.5	168.6 ± 97.9	0.12
AP threshold (mV)	-36.8 ± 3.2	-40.1 ± 2.4	0.038
AP rise time (ms)	0.53 ± 0.03	0.50 ± 0.05	0.21
AP half-width (ms)	0.92 ± 0.13	0.78 ± 0.12	0.026
AP amplitude (mV)	85.0 ± 7.8	91.4 ± 4.2	0.038
AHP amplitude (mV)	13.8 ± 1.9	15.5 ± 3.2	0.53
Repetitive firing			
Max. firing frequency (Hz)	21.7 ± 8.4	24.4 ± 6.1	0.68
Slope of F-I curve (APs/100 pA)	5.2 ± 2.6	9.4 ± 3.0	0.016
R_s (MΩ)	24.1 ± 13.3	29.1 ± 7.5	0.32

Reviewer #1:

3. The time of experiment after slicing has not been sufficiently addressed. While Fig. 2 and Fig. S2 show changes up to 3 days after resection, the implication of these results for the overall study and research question is not clear. Do the authors see the same trends/differences in the on-site tissue? More importantly, at what times were the different data points from the other figures recorded? Can the authors (statistically) exclude the possibility that the reported differences are not due to different experimental time points of on-site vs off-site recordings?

Response: Prolonged recordings from on-site tissue were not carried out in this project, so we cannot assess whether the trends or differences found in off-site tissue would also occur in on-site preparations. Except for the data now consolidated in the new Fig. 2 (previously Figs. 2 and S2), all results shown in Figs. 4–7 were obtained from slices recorded on the day of surgery (Day 1). When comparing off-site with on-site recordings, the only temporal discrepancy is a ~1.5-hour delay caused by slice handling and transport; the actual recording time points are otherwise identical. Note that comparisons between on-site and off-site electrophysiological recordings were conducted within the first 12 hours after surgery to minimize the variability in the data sets. Therefore, the observed differences are unlikely to arise primarily from variations in recording time between on-site and off-site experiments. This clarification has been added to the revised manuscript (see below).

Results, Page 10: *"Slices were recorded on the day of surgery (Day 1) either at the RWTH Aachen University Hospital (on-site) or at the Research Centre Juelich (off-site). There is an approximate 1.5-hour delay between on- and off-site recordings due to slice transportation and other handling. Note that comparisons between on-site and off-site electrophysiological recordings were conducted within the first 12 hours after surgery."*

Reviewer #1:

4. The publication that originally describes the large rhythmic depolarizations (LRD, Yang et al., Communications Biology 2024) reports different proportions of cells that are LRD+ in the pyramidal-versus the interneuron population. Within the interneuron population, there are further differences in the fraction of LRD+ cells between smaller and larger basket-cell-like cells. In the current study, the neuron type for the LRD+/LRD- neurons is not specified. The authors should thus discriminate the cell type in their analysis. Also, the authors state that "... neurons displaying spontaneous LRDs that were recorded at the off-site laboratory are located significantly deeper in the slice than those neurons showing no LRD". Are the recording depths the same on-site versus off-site?

Response: In response to the reviewer's suggestion, we now specify the neuronal cell types that exhibited LRDs: pyramidal neurons, fast-spiking (FS) interneurons, and non-fast-spiking (nFS) interneurons, for both on- and off-site recordings and for both spontaneous and NE-induced LRDs (see the new Fig. 7). In the off-site recordings, the two FS interneurons that displayed spontaneous LRDs were located deeper in the slice (85 μm and 100 μm) than FS and nFS interneurons without LRDs ($-80.1 \pm 12.8 \mu\text{m}$, $n = 12$) and pyramidal cells without LRDs ($-69.0 \pm 16.2 \mu\text{m}$, $n = 18$). Across all

datasets, recording depths for both on- and off-site experiments ranged from 50 μm to 100 μm , with an overall mean of $-73.5 \pm 15.7 \mu\text{m}$ ($n = 30$). No systematic differences in recording depth were detected between the two sites.

Minor comments:

Reviewer #1:

1. Claims in key points should be attenuated and adapted: a. First key point: The use of the word "significantly" is not ideal, as it is typically used in the context of statistical significance of a specific comparison.

Response: The word 'significantly' has now been removed.

Reviewer #1:

b. Second key point: The authors present no empirical evidence for a neuromodulator wash-out mediated effect (see comment below).

Response: To test the hypothesis that a neuromodulator washout contributes to the reduction of LRDs, we applied norepinephrine (NE, 30 μM) exogenously by bath application. As reported previously (Yang et al. Commun. Biol. 2024) and in the present study, NE induced LRDs in neurons that did not exhibit LRDs in normal artificial CSF. In addition, we performed additional experiments by perfusing human CSF into human cortical slices and found that LRD-like synaptic activity could be induced in neurons that showed no such activity in normal artificial CSF (see Fig. R3).

In summary, based on the NE and hCSF experiments, we suggest that the washout of neuromodulators during the slice handling and transport may contribute to the reduction in LRDs observed in human cortical slices.

Fig. R3. Bath-application of hCSF induced LRD-like synaptic activity in human cortical pyramidal cells. (A) Left: DIC images of the recorded human cortical neuron obtained with 4x and 40x objectives. Right: repetitive firing pattern of the neuron in response to 1-s DC current injection. (B) potential (V_m) recordings in aCSF (left) and hCSF (right). Zoom-in views of the highlighted regions are shown at the bottom.

Reviewer #1:

c. Fourth key point: It is not clear how the results of the study "advance our understanding of cortical microcircuitry" beyond the raised methodological effects.

Response: Corrected. The sentence has now been changed to "*advancing our understanding of cortical function and structure and highlighting the importance of optimizing transport protocols to preserve tissue integrity and neuronal function*".

Reviewer #1:

2. Fig S2: Could the increase in EPSP amplitude and the decrease in EPSP frequency be interrelated and biased by potential differences in noise? I assume each data point is an average from one neuron, please clarify how many events or what time period was used for the averaging. Maybe the study could benefit from controlling for detection limits as shown in this preprint: ['Mini analysis' is an unreliable reporter of synaptic changes. Ingo H. Greger, Jake F. Watson. bioRxiv 2024.10.26.620084]

Response: Spontaneous EPSPs were recorded at both on-site and off-site locations to assess the effects of tissue transport. Each data point for frequency and amplitude represents the mean value from a single neuron. In response to the reviewer's suggestion, we have added sentences to the Materials and Methods section specifying the number of events and the recording duration used for these averages (see below). The noise levels at both setups were comparable and were kept below 50 μV root-mean-square (rms). Because the amplitudes of the detected sEPSPs were typically > 200 μV —well above the noise floor. Therefore, we are confident that the electrical noise during or recordings does not have a significant impact on the the detection of spontaneous EPSPs.

Reviewer #1:

3. Fig4: How was the maximum firing frequency determined? What was the stimulation amplitude? The on-site group looks surprisingly homogeneous. Can you show an FI-slope? How do these values compare to previous reports, such as [Berg, J., Sorensen, S.A., Ting, J.T. et al. Human neocortical expansion involves glutamatergic neuron diversification. Nature 598, 151-158 (2021)] or [Planert et al., Cellular and Synaptic Diversity of Layer 2-3 Pyramidal Neurons in Human Individuals. bioRxiv 2021.11.08.467668] ?

Response: The maximum firing frequency was defined as the highest number of action potentials (APs) evoked during a 1 s DC current injection, up to the point at which repetitive firing clearly deteriorated—for example, a marked reduction in AP amplitude or a drop in firing rate caused by depolarization block. For pyramidal cells, the maximal stimulation amplitude was ~ 200 pA, whereas for fast-spiking (FS) interneurons it was ~ 300 pA. No significant difference was observed in the maximal injected current between on-site and off-site pyramidal-cell recordings (195.0 ± 54.0 pA vs. 233.3 ± 69.2 pA, $p = 0.19$).

An overlay of the F–I curves for L2/3 pyramidal cells recorded on- and off-site is shown in Fig. R4. The F–I slopes were $0.17 \pm 0.05 \text{ Hz pA}^{-1}$ for on-site and $0.14 \pm 0.05 \text{ Hz pA}^{-1}$ for off-site L2/3 pyramidal cells. These values exceed those reported in Fig. 2E of Planert et al. (bioRxiv 2021.11.08.467668), where most slopes are $\sim 0.05 \text{ Hz pA}^{-1}$. By contrast, our F–I curves closely matched those shown in Fig. 2b (bottom row) of Lee et al., Nature 598, 151–158 (2021).

Fig. R4. The relationship between firing frequency and injected current amplitude for human cortical L2/3 pyramidal cells (on-site: n = 25; off-site: n = 20) and fast-spiking interneurons (on-site: n = 14; off-site: n = 13).

Reviewer #1:

4. Fig 6: Color legend and/or schematic would help to make clear that it is PC vs FS comparison.

Response: In the new Figs. 6 and 7, a schematic for PC and FS was given for clarity. In addition, a schematic for basal, apical oblique, and apical tuft dendrites was also added to Fig. 9.

Reviewer #1:

5. Fig6: To what extent can changes on the synaptic level be explained by changes in neuronal excitability? The statement that differences are specific to interneurons is quite strong, considering potential confounds (slice variance) and low sample size. Were pyramidal neurons and interneurons from the same tissue compared?

Response: We did not find a direct relationship between changes in synaptic activity and neuronal excitability. For instance, FS interneurons recorded off-site showed a pronounced decrease in sEPSP amplitude relative to on-site recordings. By contrast, L2/3 pyramidal cells recorded off-site displayed a depolarized V_{rest} , larger AP amplitudes, and higher maximal firing frequencies, all indicative of

increased excitability. Therefore, the reduction in sEPSP amplitude observed in FS interneurons after transport is more likely attributable to altered synaptic transmission—such as a lower release probability or a reduced number of release sites—than to changes in intrinsic excitability.

Reviewer #1:

6. P20: Here it says "off-site" twice: "... the amplitude of EPSPs in interneurons was significantly reduced when recordings are conducted off-site compared to off-site recordings."

Response: corrected.

Reviewer #1:

7. Fig 7C typo (p=p=).

Response: corrected.

Reviewer #1:

8. Fig7C and 7E: "n.s. $p > 0.05$ " is bit confusing, suggesting that the comparisons were not significant while they are. Maybe rephrase.

Response: corrected.

Reviewer #1:

9. Fig7C reports 2/67 neurons for the off-site group. However, the text states $n=2$ for LRD+ and $n=30$ for LRD- neurons (page 21). This difference in "n's" should be resolved.

Response: In total, 67 neurons recorded off-site were examined for the presence of LRDs. However, somatic depth was recorded for only 30 out of these neurons. Therefore, the number of neurons reported in the text is 30, rather than 67 as shown in the figure. This clarification has been added to the revised manuscript.

Reviewer #1:

10. P21: Spontaneous LRD are found in "significantly" deeper neurons. "Significantly" is a strong word considering that only $n = 2$ of neurons with LRD+ were recorded. Is there a statistical confirmation? Was soma depth also predictive for NE-responsiveness?

Response: Agreed. We removed the word 'significantly' from the sentence to avoid exaggeration. We did not find any relationship between soma depth and NE-responsiveness in neurons recorded at either on- or off-site locations.

Reviewer #1:

11. Fig 8 and Table 3: A visualization of the quantification in the figure (bar plots with scatter) could help convey the message and be more consistent with other results.

Response: The new Fig. 8B and 8D were created to include a visualization of the morphological quantification.

Reviewer #1:

12. Fig 9: Similar to general comment to other figures: From how many samples were the neurons taken? Does n represent region/dendrite imaged or individual neurons? From how many surgeries were these data collected?

Response: In general, n represents individual neurons from which the parameters were extracted. The number of surgeries from which each neuron was recorded has now been added to the figure legends.

Reviewer #1:

13. P25: "comprehensive" is quite a strong word for n = 10 in each condition.

Response: In the revised manuscript, the number of neurons recorded has been substantially increased as mentioned above. Furthermore, this manuscript characterizes multiple properties – including intrinsic membrane properties, spontaneous excitatory synaptic activities, dendritic morphologies, and spine densities - to demonstrate the effects of transport on both human cortical pyramidal cells and fast spiking interneurons. Therefore, we think our work is a 'systematic' study. We replaced 'comprehensive' with 'systematic' in the manuscript now.

Reviewer #1:

14. P26: If reduction in LRDs is "most noteworthy", consider expanding on this result a bit more. What about amplitude and frequency of LRD? If LRDs are predominantly reported in interneurons, how many of the data in Fig7C were from interneurons and how many were PCs?

Response: We have added sentences describing the amplitude and frequency of LRDs recorded at both on- and off-sites (see below). No significant differences were observed in either amplitude (9.0 ± 3.5 mV vs. 6.4 ± 2.7 mV, $p = 0.15$) or frequency (0.31 ± 0.18 Hz vs. 0.34 ± 0.17 Hz, $p = 0.61$) when comparing on-site to off-site recordings. Similarly, no significant differences were found both in amplitude (8.7 ± 4.2 mV vs. 9.6 ± 1.4 mV, $p = 0.39$) or frequency (0.35 ± 0.19 Hz vs. 0.24 ± 0.15 Hz, $p = 0.22$) when comparing spontaneous LRDs to NE-induced LRDs. The only notable difference is the percentage of recorded neurons exhibiting LRDs, as shown in Fig. 7C. In the new Fig. 7, the subtypes of neurons showing LRDs recorded at both on- and off-sites are now given.

Results, Page 14: *“In addition to comparing the incidence of LRDs between on- and off-site recordings, the amplitude and frequency of LRDs were analyzed. No significant differences were observed in amplitude (9.0 ± 3.5 mV vs. 6.4 ± 2.7 mV, $p = 0.15$) or frequency (0.31 ± 0.18 Hz vs. 0.34 ± 0.17 Hz, $p = 0.61$) between on- and off-site recorded LRDs. Similarly, no significant differences were found in amplitude (8.7 ± 4.2 mV vs. 9.6 ± 1.4 mV, $p = 0.39$) or frequency (0.35 ± 0.19 Hz vs. 0.24 ± 0.15 Hz, $p = 0.22$) when comparing spontaneously generated LRDs to NE-induced LRDs.”*

Reviewer #1:

15. P26: Why is the reduction in LRD due to neuromodulator wash-out? Could the reduction not be well in line with the detected synaptic and spine changes? Overall, I do not see much support for the proposed neuromodulator wash-out hypothesis. It is likely that a few minutes of perfusion in the recording chamber would already substitute most of compounds in the extracellular solution, considering the kinetics of typical pharmacological wash-in and wash-out experiments. Furthermore, the result in Fig. 7 suggests rather the opposite: NE condition increased LRD in both conditions by roughly 200%. If the transport-related changes would be due to neuromodulator wash-out, I would expect that NE should have a differential and rescuing effect on the off-site slices.

Response: We agree with the reviewer that we do not have proof of the mechanism underlying the transport-related effects on the LRD generation described in this study. The reduction in the occurrence of LRDs could be due to multiple factors. By analyzing the intrinsic membrane properties, spontaneous excitatory synaptic activities, dendritic morphologies, and spine densities, we found a decrease in the sEPSP amplitudes in FS interneurons that might be in part responsible. The reduction in pyramidal cell spine densities could be another contributing factor. Washout of neuromodulators such as acetylcholine, norepinephrine (NE), dopamine, and others may also play a critical role. To test this in principle, we used exogenous bath-application of NE (30 μ M) to induce LRD in neurons that did not display LRDs in normal artificial CSF, as reported previously (Yang et al. Commun. Biol. 2024) and in the present study. To address the potential mechanism, we have revised this in our discussion.

Discussion, Page 17: *“As LRDs are predominantly observed in human interneurons rather than in pyramidal cells and depend on the glutamatergic transmission (Yang et al., 2024), the decreased EPSP amplitude found in off-site recorded interneurons may serve as an indicator of reduced LRD occurrence. The reduction in pyramidal cell spine densities could be another contributing factor. This is supported by the fact that transport to an off-site laboratory has an adverse impact on network-level events, possibly due to subtle changes in the synaptic connectivity and neuromodulatory environment of neurons. It may be hypothesized that vibrations occurring during transport could therefore diminish the concentrations of neuromodulators, such as NE, in the intercellular space. This could, in turn, lead to a substantial influence on the occurrence of network events such as LRDs (Yang et al., 2024). To test this in principle, we used exogenous bath-application of NE (30 μ M) to induce LRD in neurons that did not display LRDs in normal artificial CSF, as reported previously (Yang et al. Commun. Biol. 2024) and in the present study.”*

Reviewer #1:

16. If the authors have included data from later days and considering the non-sterile conditions, it is likely that bacterial growth has occurred that could have activated microglia. While microglia-activation has been briefly mentioned for the discussion of intrinsic properties, one could also hypothesize that most of the observed effects could arise from microglia mediated changes, for example due their established role in synaptic pruning. Because it is well established that microglia are activated by physical stress (and infection), the discussion on potential mechanisms and improvement should elaborate more on these aspects. The benefits of sterile preparation, transport and slice storage conditions as well as potential improvements to reduce mechanical stress during transport should be discussed.

Response: We thank the reviewer for the insightful comment. As noted in our response above, except for the data in Fig. 2 and Fig. S2 (now combined into the new Fig. 2), all other data in Figs. 4- 7 were obtained from recordings performed on the day of neurosurgery (Day 1, within 12 hours). We have now expanded the discussion to address the potential effects of microglia and added possible improvements, such as maintaining sterile conditions during preparation, transport, and slice storage, as well as reducing mechanical stress by using an anti-vibration box to stabilize the slice keeper during transport (see below).

Discussion, Page 19: *“Because microglia are well known to be activated by physical stress and infection (Avata and Schäfer, 2020; Quan and Zhang 2023), we hypothesize that many of the observed effects could result from microglia-mediated changes, such as their established roles in synaptic pruning (Paolicelli et al., 2011; Schafer et al., 2012; Gunner et al.; 2019). Therefore, potential improvements - such as using an anti-vibration box to store the slice holder during transport to reduce mechanical stress, and maintaining sterile conditions during slice preparation, transport, and storage – would likely be beneficial.”*

Reviewer #2:

Single cell electrophysiology of human cortical neurons is rare and thus very valuable. However, at present, the data do not convincingly support the result, that differences measured on-site vs off-site are really due to the transport. As patch-clamp recordings usually follow a tightly controlled experimental design, a data set of $n=10$ or 11 should be sufficient if recorded at the same setup by the same experimenter. Since this is not the case in your study, you would 1) need a significantly larger data set, 2) indicate pairs of neurons that originate from the same probe (link the dots in the graphs by lines or color code the tissue samples), and 3) perform paired statistics. Comparing data from the same probe is of outmost importance since I suppose neurons from epilepsy patients maybe more excitable than those of tumor patients and could induce bias independent of transport. Once color/symbol coded, you could state in the figure legends from which sample the neurons were recorded.

Response: We greatly appreciate the reviewer's thorough assessment and constructive feedback. In the revised manuscript, we have substantially increased our sample sizes by re-analyzing existing patch-clamp data, performing additional electrophysiological recordings, and adding further morphological reconstructions. As mentioned in our reply to reviewer 1, in the electrophysiological analysis of human L2/3 pyramidal cells (Fig. 4), the number of neurons (n) was increased from $n = 10$ to 25 for on-site recordings and from $n = 11$ to 20 for off-site recordings. Similarly, the sample sizes for Figs. 5, 6, 8, and 9 have all been increased to ≥ 10 .

For selected data sets—such as the electrophysiology of human L2/3 pyramidal cells—we carried out paired statistical analyses (Fig. R1). These analyses revealed significant differences in parameters such as V_{rest} and maximal firing frequency. Other parameters, including rheobase current and action-potential amplitude, exhibited trends but did not reach statistical significance, likely owing to the still-limited sample sizes.

We also compared human L2/3 pyramidal cells recorded on- and off-site between epilepsy and tumor cases (Fig. R5). No clear differences were detected between the two patient groups for either recording-site condition. Consequently, data from both groups were pooled in the present study.

Fig. R1. Paired comparison of electrophysiological properties in human cortical L2/3 pyramidal cells recorded on- and off-site. Data points were calculated by averaging the parameters extracted from individual PCs obtained from the same surgery. In total, parallel on- and off-site recordings were obtained from six surgeries, yielding 20 PCs recorded on-site and 16 PCs recorded off-site. P values were calculated using the Wilcoxon signed-rank test for paired data.

Fig. R5. Comparison of electrophysiological properties of human L2/3 PCs recorded on-site and off-site in cortical slices from epilepsy (E) and tumor (T) patients. Data points were grouped into four categories: on-site (E, n = 18), on-site (T, n = 7), off-site (E, n = 12), and off-site (T, n = 8). P values shown above each sub-figure were calculated using the non-parametric Kruskal-Wallis test for multiple comparisons. Pairwise p values were calculated using the Wilcoxon-Mann-Whitney test.

Reviewer #2:

I suggest the following experiment to control for changes due to transport (independent of confounding factors such as a second setup or another experimenter). You treat a batch of tissue samples exactly as if transporting them to the off-site location, but then drive around for the same time and return to the on-site location where the same setup and same experimenter will perform the "post-transport" experiments.

Response: Following the reviewer's suggestion additional experiments were conducted on human slices obtained from three surgeries, with or without a round transport (30-40 minutes), at the on-site laboratory. As shown in Fig. R2 and Table R1, comparison of transported versus non-transported slices revealed statistically significant changes in V_{rest} , the action potential (AP) half-width, and AP amplitude, which are consistent with the findings in Fig. 4B. In contrast, no significant differences were observed in sEPSP frequency or amplitude, consistent with the results shown in Fig. 6B. These

Fig. R2. Comparison of the intrinsic and synaptic properties of human cortical L2/3 pyramidal cells recorded in control and transported slices at the same setup (Aachen) by the same experimenter (GQ). (A) Comparison of electrophysiological properties between neurons recorded in control (n = 7) and transported (n = 7) slices. (B) Comparison of spontaneous EPSP properties between neurons recorded in control (n = 7) and transported (n = 7) slices. P values were calculated using the Wilcoxon-Mann-Whitney two-sample rank test.

results demonstrate that the observed transport effects cannot be attributed to confounding factors such as differences in experimental setup or experimenter (see also our reply to reviewer #1).

Reviewer #2:

An important take-home message from your study also outside possible medical applications is the fact that you can keep brain slices alive and healthy for up to 60 hours and share them between reasonable locations. This is an important point to make as it will be a revolution regarding reducing animal numbers in research (3Rs) and scientist could share rodent brain slices across campus or even further.

Response: In response to the reviewer's suggestion, additional experiments were performed in which rodent brain slices were recorded at the Jülich laboratory (by DY) as well as transported to and recorded at the Aachen laboratory (by GQ). As shown in Fig. R6, transport of rat and mouse cortical slices is feasible, and recordings from various neuron types can be obtained after transport. Nevertheless, pronounced alterations were observed: for example, the excitatory-to-inhibitory

Fig. R6. Patch-clamp recordings from transported rat and mouse cortical slices. (A) Patch-clamp recordings were performed in transported cortical slices from the adult (>P30) rat ($n = 5$ rats) medial prefrontal cortex (mPFC). Representative firing patterns are shown for three classical neuronal subtypes: pyramidal cells, fast-spiking (FS), and non-fast-spiking (nFS) interneurons. (B) Composition of neuronal subtypes recorded in control (on-site) and transported (off-site) adult rat mPFC slices. Note that the ratio of recorded excitatory and inhibitory neurons changed after transport, from 3.5:1 in on-site slices to 0.9:1 in off-site slices. (C) No significant changes were observed in intrinsic properties (e.g., resting membrane potential, input resistance, AP half-width) or synaptic properties (sEPSP amplitude and frequency, not shown) of excitatory neurons recorded in off-site ($n = 8$ neurons) versus on-site slices ($n = 20$ neurons). (D) Patch-clamp recordings were also performed in transported cortical slices from young (P14) mouse hippocampal area CA2. Pyramidal cells exhibited both regular spiking and rhythmic bursting firing patterns, similar to recordings from control (non-transported) slices (Qi et al. *Brain*, awaf243; <https://doi.org/10.1093/brain/awaf243>).

neuron ratio shifted from 4:1 in control slices to 1:1.1 in transported slices. Moreover, unlike human cortical tissue, rodent slices stored overnight (Day 2) showed a marked decline in quality, and no healthy neurons could be recorded, regardless of transport. These results suggest that human cortical slices are uniquely robust compared with rodent slices. While this data is very interesting, we decided not to include this in the manuscript due to the limited scope of the present study.

Reviewer #2:

I would also like to add a few minor comments.

In your key points, rather than stating "changes of neuronal excitability", please state whether neuronal excitability increase or decreases.

Response: Corrected, phrase was changed to 'an increased neuronal excitability'.

Reviewer #2: The change in neuromodulator action has not been tested in your experiments, but is a discussion point. Though a valid discussion point, it is not one of your key findings. Narrower action potentials could for example also be caused by slight differences in the recording temperature between the two setups.

Response: We agree with the reviewer and have modified our statements and strengthened our discussion to also discuss other potential alterations underlying the observed effects.

For example: "The decreased EPSP amplitude could reflect reduced presynaptic release probability or postsynaptic receptor sensitivity. Recent evidence also suggests that human pyramidal neurons may exhibit enhanced NMDA receptor contributions to synaptic transmission compared to rodent neurons (Hunt, 2023; Eyal, 2018), potentially rendering these synapses more vulnerable to metabolic or ionic perturbations during transport, though this remains to be directly tested."

To address temperature or other differences between the setups, we added a table and a detailed description of the recording settings:

Methods, Page 6: *"The setups used for on- and off-site recordings were nearly identical, with only minor differences, such as the light microscopy used for imaging (Table 2). Furthermore, the experimenters for on-site (DY) and off-site (GQ) recordings were trained in the same laboratory using the same patch-clamp setup. These efforts were taken to reduce bias related to the setup or the experimenter in the recording procedures."*

Results Page 12: *"An objective comparison of on-site and off-site data may be influenced by systematic bias introduced using different setups or different experimenters. To minimize the potential for such bias, we used experimental setups at both sites that were very similar in composition (see Table 2). Additionally, the experimenters who conducted patch-clamp recordings at both on-site and off-site locations had very similar technical expertise. To further minimize bias related to setups and experimenters, all experiments were performed by the same experimenter using human cortical slices obtained from three surgeries. These experiments were conducted either directly in the on-site laboratory or after a 30–40-minute round-trip transport."*

Reviewer #2: Key points 3 and 4 are again conclusions rather than results of your study. I agree it is an important point to make, but these two points should be combined so that out of the 4 key points are at least three that state your findings.

Response: Corrected, key points 3 and 4 are now combined into key point 3, see below.

On page 2, “Advancing Understanding of Cortical Function and Structure: This research provides valuable insights into the impact of transportation on human brain tissue, advancing our understanding of cortical function and structure and highlighting the importance of optimizing transport protocols to preserve tissue integrity and neuronal function.”

Reviewer #2: Your rationale of using box plots for $n > 10$ and bar graphs for $n < 10$ is uncommon and not consistent in your figures (for example fig 6B should be bar graphs according to your rule). Usually bar graphs are used when data are not normally distributed, while bar graphs are suited for normally distributed data. As you used non-parametric statistics, I recommend to use bar graphs in all your figures, instead of bar graphs.

Response: By re-analyzing existing patch-clamp data, performing additional electrophysiological recordings and morphological reconstructions, numbers for each dataset were increased to ≥ 10 . Therefore, all figures now include box plots and not bar graphs.

Reviewer #2: In your figure 5B, you report the off-site APs to be briefer but larger in amplitude. Independent of where or when APs were recorded, smaller AP half-width usually is due to stronger contribution of high-voltage gated potassium currents, but if that is the case, I would expect the AP amplitudes to be the same or smaller rather than larger. Perhaps in your case the change in half-width is due to a change in AP rise time rather than repolarization. You could provide these measures for the data you have and thereby hint at possible mechanisms.

Response: Following the reviewer’s suggestion, AP rise time has been analyzed and added to Table 2. Here, AP rise time is defined as the interval between the AP threshold and the AP peak. However, differences in AP rise time between neurons recorded on- and off-site did not reach statistical significance for either pyramidal cells (PCs) or FS interneurons. This suggests that the differences observed in AP half-width for both PCs and FS interneurons may be related to the repolarization phase, potentially involving potassium conductances. In addition to AP rise time, the AP decay time, defined as the interval between the AP peak and the AHP trough, was analyzed. We indeed found statistically significant differences in the AP fall time for both pyramidal cells (3.07 ± 0.98 ms, $n = 25$ for on-site vs. 2.47 ± 0.57 ms, $n = 20$ for off-site; $p = 0.034$) and FS interneurons (1.30 ± 0.68 ms, $n = 14$ for on-site vs. 0.78 ± 0.27 ms, $n = 13$ for off-site; $p = 0.022$).

Reviewer #2: In general, rather than making a strong point about the transport, maybe highlight, the variability that can occur do to different factors such as transport, setups, experimenter, time post tissue harvest etc.

Response: We have now added sentences in the Discussion to highlight the variability that can occur due to different factors such as transport, setups, experiments, and time post tissue harvest below. Importantly, our newly added experiments, performed with the same setup and the same experimenter, still revealed clear differences between transported and non-transported slices (Fig. R2, Table R1). This indicates that transport per se, independent of confounding factors such as setup or experimenter, can indeed affect neuronal properties.

Discussion, Page 18: *“It should be noted that, in addition to transport itself, variability in slice preparation, transport, experimental setup, experimenter, and time post-harvest may also influence the final results.”*

Dear Dr Koch,

Re: JP-RP-2025-288111R1 "Transport-Related Effects on Intrinsic and Synaptic Properties of Human Cortical Neurons: A Comparative Study" by Guanxiao Qi, Danqing Yang, Aniella Vanessa Bak, Werner Hucko, Daniel Delev, Hussam Aldin Hamou, Dirk Feldmeyer, and Henner Koch

Thank you for submitting your manuscript to The Journal of Physiology. It has been assessed by a Reviewing Editor and by 2 expert referees and we are pleased to tell you that it is acceptable for publication following satisfactory revision.

REVISION CHECKLIST:

Please upload two versions of your manuscript text: one with all relevant changes highlighted and one clean version with no changes tracked. The manuscript file should include all tables and figure legends, but each figure/graph should be uploaded as separate, high-resolution files. The journal is now integrated with Wiley's Image Checking service. For further details, see: <https://www.wiley.com/en-us/network/publishing/research-publishing/trending-stories/upholding-image-integrity-wileys->

image-screening-service

We look forward to receiving your revised submission.

Yours sincerely,

Katalin Toth
Senior Editor
The Journal of Physiology

REQUIRED ITEMS

(1) You must start the Methods section with a paragraph headed Ethical Approval. If experiments were conducted on humans, confirmation that informed consent was obtained, preferably in writing, that the studies conformed to the standards set by the latest revision of the Declaration of Helsinki and that the procedures were approved by a properly constituted ethics committee, which should be named, must be included in the article file. If the research study was registered (clause 35 of the Declaration of Helsinki), the registration database should be indicated, otherwise the lack of registration should be noted as an exception (e.g. The study conformed to the standards set by the Declaration of Helsinki, except for registration in a database). For further information see: <https://physoc.onlinelibrary.wiley.com/hub/human-experiments>.

(2) A Data Availability Statement is required for all papers reporting original data. This must be in the Additional Information section of the manuscript itself. It must have the paragraph heading 'Data Availability Statement'. All data supporting the results in the paper must be either: in the paper itself; uploaded as Supporting Information for Online Publication; or archived in an appropriate public repository. The statement needs to describe the availability or the absence of shared data. Authors must include in their statement: a link to the repository they have used, or a statement that it is available as Supporting Information; reference the data in the appropriate sections(s) of their manuscript; and cite the data they have shared in the References section. Whenever possible, the scripts and other artefacts used to generate the analyses presented in the paper should also be publicly archived. If sharing data compromises ethical standards or legal requirements then authors are not expected to share it, but must note this in their statement. For more information, see our Statistics Policy.

(3) Please include an Abstract Figure file and an Abstract Figure legend (**we seem to be missing the legend**). An appropriate figure legend, which should not exceed 150 words in length, should be included in the main manuscript file. The Abstract Figure is a piece of artwork designed to give readers an immediate understanding of the research and should summarise the main conclusions. If possible, the image should be easily 'readable' from left to right or top to bottom. It should show the physiological relevance of the manuscript so readers can assess the importance and content of its findings. Abstract Figures should not merely recapitulate other figures in the manuscript. Please try to keep the diagram as simple as possible and without superfluous information that may distract from the main conclusion(s). Abstract Figures must be provided by authors no later than the revised manuscript stage and should be uploaded as a separate file during online submission labelled as File Type 'Abstract Figure'. Please also ensure that you include the figure legend in the main article file. All Abstract Figures should be created using BioRender. Authors should use The Journal's premium BioRender account to export high-resolution images. Details on how to use and access the premium account are included as part of this email.

EDITOR COMMENTS

Reviewing Editor:

Thank you for your thorough revision and addressing the reviewers' comments. Both reviewers are satisfied by your revisions, so that your manuscript should now be publishable in the Journal of Physiology.

Senior Editor:

Before final acceptance, please add:

-mention of Declaration of Helsinki

-data availability statement

-abstract figure legend

REFeree COMMENTS

Referee #1:

See attached Word file.

Referee #2:

The manuscript by Qi and colleagues provides a substantial, very valuable data set of electrophysiological and anatomical data of human cortical pyramidal neurons and fast firing interneurons, recorded and compared across two test sites. All prior concerns were addressed with care. I recommend the manuscript for publication in the Journal of Physiology.

END OF COMMENTS

1. Overall, while the study identifies parameters that differ between on-site and off-site recordings, the narrative seems to be too focused on these changes and misses the opportunity to highlight the notable robustness of many key electrophysiological properties (e.g., AP kinetics, sEPSP activity). I feel that establishing the stability of these parameters after transport are at least equally important for the community. Below are some comments on the specific results:

a. Focus on LME, paired surgery test and transport control: When comparing neuronal properties, the authors report four different statistical results: Wilcoxon-Mann-Whitney test at single neuron level and a LME, paired and averaged surgery level and paired recordings from the same setup (with only transport difference). I have tried to summarize the reported results in the following table:

	Wilcoxon-Mann-Whitney (n = 25 vs n = 20)	LME(n = 20 vs n = 16, surgeries = ?)	Paired surgery average (n = 6 surgeries)	Same setup, only transport, (n = 10 vs n = 10 neurons, 3 surgeries)
Vrest	0.001	0.00001	0.03	0.03
Tau time constant	0.002	0.0008	?	?
Rheobase	0.006	0.004	0.09	0.12
AP half width	0.045	0.22	0.56	0.03
AP amplitude	0.01	0.01	0.09	0.04
max frequency	0.00002	0.00002	0.03	0.7

No significant differences: Rin, Sag, AP threshold, AP rise time, AHP amplitude, FI slope, sEPSP. While the paired surgery averages and control experiments with only transport provide important additional information, I feel that reporting the p-values of both Wilcoxon-Mann-Whitney test and from the LME appears redundant. Especially, as the LME is much more applicable to this situation, I would suggest only including the p-values from the LME. For example, table 1 could include the p-values of LME, paired surgery, only transport instead of the Wilcoxon-Mann-Whitney test.

b. Effect on rheobase is not so clear: The difference in rheobase, and hence change in excitability, was only significant in the LME, but did not hold at the paired surgery level and also was not evident when only transport conditions were compared. I would suggest attenuating the claim when reporting it in the manuscript (e.g. abstract: “off-site ... showed ... significantly lower rheobase current”). Also, given the additional transport control experiment, I don’t think that this statement is sufficiently supported by the data: “specific intrinsic properties, such as resting membrane potential and rheobase

current, are influenced by the transportation process.” Also, discussion should focus more on the more robust result of depolarized neurons, rather than the less clear effect on rheobase.

- c. Highlight and discuss stable parameters: When summarizing these results on pyramidal cells, I think the unchanged parameters are of equal importance. The community would benefit from a more elaborate discussion of the evidence that many properties remained stable after transport, supporting the validity of this method. I suggest to adapt keypoint 1 to reflect this and provide a more nuanced discussion on differences that appear to be also cell type specific (RMP only in PC, sEPSP only in interneurons). In manuscript elaborate on unchanged “overall firing capability”.
 - d. Overinterpretation of sEPSP results: The discussion states that “synaptic function is particularly vulnerable to transport-related effects,” yet only one of four assessed EPSP parameters shows a significant difference. This interpretation appears overstated. The data instead suggest that synaptic function in pyramidal neurons remains stable, with only the large-amplitude EPSPs onto interneurons (“VLEs”; Szegedi et al., PLOS Biology 2016) showing sensitivity to transport. The conclusion should be revised to reflect this more limited and cell-type-specific effect.
2. The authors have performed additional experiments and analyses on the LRD which clarified several points raised. However, some aspects with respect to sampling confounds and interpretation remain unclear:
- a. Although neuron types (FS, nFS, PC) are now shown for cells exhibiting LRDs, the same distinction is missing for cells without LRDs. Given known subtype-specific differences in LRD activity (Yang et al., 2024), it remains unclear whether the altered off-site LRD fractions reflect true effects or simply differences in subtype sampling.
 - b. The reduced NE responsiveness in off-site recordings does not clearly support a specific “neuromodulator wash-out hypothesis,” as many other mechanisms could underlie a decreased sensitivity to NE of neurons. Unless more direct evidence is provided, or a direct link to the human CSF experiments explained, the interpretation should be reframed more broadly as altered neuromodulator signaling rather than a specific washout effect.
 - c. Is there a quantification or visualization backing this statement: “This induces LRD-like synaptic activity in neurons that did not exhibit LRD-like activity in artificial CSF.”

Minor:

1. The same statistical approach used for the pyramidal cells should also be applied when comparing the FS interneurons. If the data on paired surgery level or transport only are not available, then at least the LME results should be reported and the lack of other controls discussed.

2. The statement that ICC values of 0.3 or 0.16 indicate that differences in Vrest and FI-slope were “likely due to the surgeries” is somewhat misleading. These ICCs simply reflect how much variance is attributable to surgery versus recording location, not that surgery is the primary cause of the observed differences.
3. “The findings of this study highlight how transport-induced stress can lead to significant alterations in synaptic efficacy and network dynamics, notably the reduction in EPSP amplitudes and LRDs.” Please specify “reduction in EPSP amplitudes of FS interneurons”
4. The lack of difference in RMP in FS interneurons is interesting, since this was so striking in the pyramidal cell dataset. Could this be mentioned and potentially discussed?
5. Fig 7 / LRD: It appears that the authors used a one-tailed chi-square test. Since chi-square tests do not generally test a directional hypothesis and given the small number of observations, a two-tailed Fishers exact test would be more appropriate. The results of the chi-square test or Fisher exact test should also be referenced in the manuscript.
6. If n numbers are provided within the abstract (n = 200 neurons from 32 surgeries), then it should also or instead state the number of neurons that passed quality control and went into analyses.
7. The discussion reads: “The mechanical stress encountered during transport can alter these ECM components, leading to changes in tissue hydration, viscoelastic properties, and membrane fluidity, ultimately affecting cellular behaviors such as signaling pathways.” As this is a speculation, I would suggest rephrasing “can” to “could”.

NOTE: The comments from Editor and Reviewers are in black and the responses from Authors are in blue.

EDITOR COMMENTS

Reviewing Editor:

Thank you for your thorough revision and addressing the reviewers' comments. Both reviewers are satisfied by your revisions, so that your manuscript should now be publishable in the Journal of Physiology.

Senior Editor:

Before final acceptance, please add:

-mention of Declaration of Helsinki

-data availability statement

-abstract figure legend

Response to the Editor's Comments: We thank the editor for the appreciation of our work. In the revised manuscript, we supplied the required items, see below:

- (1) In the Ethical Approval, the Declaration of Helsinki was mentioned (on Page 5).
- (2) A Data availability statement was added (on Page 20).
- (3) A legend was added to the Abstract Figure.

Reply to Reviewer 1’s further comments:

- Overall, while the study identifies parameters that differ between on-site and off-site recordings, the narrative seems to be too focused on these changes and misses the opportunity to highlight the notable robustness of many key electrophysiological properties (e.g., AP kinetics, sEPSP activity). I feel that establishing the stability of these parameters after transport are at least equally important for the community. Below are some comments on the specific results:

Response: We agree with your point that highlighting the robustness of many key electrophysiological and morphological properties (e.g., the overall dendritic architecture) after transport is equally important and necessary. In line with your comments, we have revised the manuscript as follows:

In the Abstract, we wrote: ‘Action potential (AP) firing patterns remained largely preserved across both recording sites, but several differences were observed.’ and ‘Although overall dendritic architecture was preserved...’

In the Discussion, in the 1st paragraph, the following sentences were added: ‘Our results indicate that neurons in brain slices transported for 30-40 minutes from the surgical resection site to an off-site laboratory retain notable robustness of many key electrophysiological (e.g., AP kinetics, sEPSP activity) and morphological (e.g., overall dendritic architecture) properties, while still exhibiting measurable alterations in certain intrinsic, synaptic, and morphological properties.’

- Focus on LME, paired surgery test and transport control: When comparing neuronal properties, the authors report four different statistical results: Wilcoxon-Mann-Whitney test at single neuron level and a LME, paired and averaged surgery level and paired recordings from the same setup (with only transport difference). I have tried to summarize the reported results in the following table:

	Wilcoxon-Mann-Whitney (n = 25 vs n = 20)	LME(n = 20 vs n = 16, surgeries = ?)	Paired surgery average (n = 6 surgeries)	Same setup, only transport, (n = 10 vs n = 10 neurons, 3 surgeries)
Vrest	0.001	0.00001	0.03	0.03
Tau time constant	0.002	0.0008	?	?
Rheobase	0.006	0.004	0.09	0.12
AP half width	0.045	0.22	0.56	0.03
AP amplitude	0.01	0.01	0.09	0.04
max frequency	0.00002	0.00002	0.03	0.7

No significant differences: Rin, Sag, AP threshold, AP rise time, AHP amplitude, FI slope,

sEPSP. While the paired surgery averages and control experiments with only transport provide important additional information, I feel that reporting the p-values of both Wilcoxon-Mann-Whitney test and from the LME appears redundant. Especially, as the LME is much more applicable to this situation, I would suggest only including the p-values from the LME. For example, table 1 could include the p-values of LME, paired surgery, only transport instead of the Wilcoxon-Mann-Whitney test.

Response: We agree that the LME is more applicable to the situation where more than one neuron (e.g., $n = 3-5$ or more) was collected from an individual surgery. This is the case for the L2/3 pyramidal cell data in this study. Therefore, we added a paragraph to address this point in the last version of the revised manuscript. In the latest version of our manuscript, the LME analysis was also performed for L2/3 FS interneurons, see our response to Minor Comment 1 below. However, for other analyses throughout the manuscript, such as the sEPSP, the dendritic morphology, and spine density, in most cases only a single neuron was collected per surgery. Therefore, the LME may not be necessary or applicable to them. Instead, the Wilcoxon-Mann-Whitney test is more appropriate. For consistency, we prefer to retain the p-values calculated using the Wilcoxon-Mann-Whitney test in Table 3. Instead, a new table to summarize all the statistics, like the one shown above, was added as the Statistical Summary Document.

- b. Effect on rheobase is not so clear: The difference in rheobase, and hence change in excitability, was only significant in the LME, but did not hold at the paired surgery level and also was not evident when only transport conditions were compared. I would suggest attenuating claim when reporting it in the manuscript (e.g. abstract: “off-site ... showed ... significantly lower rheobase current”). Also, given the additional transport control experiment, I don’t think that this statement is sufficiently supported by the data: “specific intrinsic properties, such as resting membrane potential and rheobase current, are influenced by the transportation process.” Also, discussion should focus more on the more robust result of depolarized neurons, rather than the less clear effect on rheobase.

Response: We revised the manuscript to weaken our statements of the finding of a lowered rheobase current in off-site recorded pyramidal cells in line with your comments, see below:

In the Abstract, ‘significantly’ was deleted from the sentence: ‘Off-site recorded pyramidal cells showed a depolarized resting membrane potential and a lowered rheobase current.’

In Results, page 12, ‘rheobase current’ was removed from the sentence: ‘specific intrinsic properties, such as resting membrane potential, AP half-width and AP amplitude, are influenced by the transportation process.’

- c. Highlight and discuss stable parameters: When summarizing these results on pyramidal cells, I think the unchanged parameters are of equal importance. The community would benefit from a more elaborate discussion of the evidence that many properties remained stable after transport, supporting the validity of this method. I suggest to adapt keypoint 1 to reflect this and provide a more nuanced discussion on differences

that appear to be also cell type specific (RMP only in PC, sEPSP only in interneurons). In manuscript elaborate on unchanged “overall firing capability”.

Response: The key point 1 has been revised following your suggestion, see below:

‘Effects of Transportation on Neuronal Properties: Brief transportation of human brain tissue retains many key neuronal properties, while still exhibiting measurable alterations in certain intrinsic, synaptic, and morphological properties.’

In the Discussion, 1st paragraph, a sentence was added: ‘These alterations are cell-type specific, e.g., resting membrane potential changes only in pyramidal cells and sEPSP amplitude changes only in FS interneurons.’

- d. Overinterpretation of sEPSP results: The discussion states that “synaptic function is particularly vulnerable to transport-related effects,” yet only one of four assessed EPSP parameters shows a significant difference. This interpretation appears overstated. The data instead suggest that synaptic function in pyramidal neurons remains stable, with only the large-amplitude EPSPs onto interneurons (“VLEs”; Szegedi et al., PLOS Biology 2016) showing sensitivity to transport. The conclusion should be revised to reflect this more limited and cell-type–specific effect.

Response: The conclusion on this point has been revised, see below:

In Discussion, page 17, ‘Our analysis of spontaneous excitatory postsynaptic potentials (EPSPs) indicates that transport-related effects on synaptic function are cell-type dependent.’

And ‘While the frequency of spontaneous EPSPs did not differ significantly between on-site and off-site recordings for both pyramidal cells and interneurons, there was a notable reduction in EPSP amplitudes in off-site recorded interneurons due to the reduction of large-amplitude EPSPs.’

2. The authors have performed additional experiments and analyses on the LRD which clarified several points raised. However, some aspects with respect to sampling confounds and interpretation remain unclear:

- a. Although neuron types (FS, nFS, PC) are now shown for cells exhibiting LRDs, the same distinction is missing for cells without LRDs. Given known subtype-specific differences in LRD activity (Yang et al., 2024), it remains unclear whether the altered off-site LRD fractions reflect true effects or simply differences in subtype sampling.

Response: To exclude the effects from the differences in subtype sampling, we revised the part for LRD in the manuscript, see below:

On pages 13 and 14, the text reads now: ‘At the on-site laboratory, 10% of the recorded neurons (n = 133 including 64 pyramidal cells and 69 interneurons) exhibited spontaneous LRDs, whereas only 3% of the neurons recorded at the off-site laboratory (n = 67 including 29 pyramidal cells and 38 interneurons) displayed spontaneous LRDs (Fig. 7A-C).’

Compared to sampling for on-site recordings (the ratio is 1:1 for pyramidal cells vs. interneurons), more interneurons were sampled for off-site recordings compared to pyramidal cells, i.e., the ratio is 1.3:1. However, there are still rare neurons exhibiting LRDs that were recorded at off-site.

- b. The reduced NE responsiveness in off-site recordings does not clearly support a specific “neuromodulator wash-out hypothesis,” as many other mechanisms could underlie a decreased sensitivity to NE of neurons. Unless more direct evidence is provided, or a direct link to the human CSF experiments explained, the interpretation should be reframed more broadly as altered neuromodulator signaling rather than a specific washout effect.

Response: We agree with this point and made changes in the manuscript accordingly, see below:

Key point 2 now reads: ‘Mechanical Stress and Neuromodulator Dysfunction May Underlie Alterations: These changes are likely due to the combined effects of mechanical stress and altered neuromodulator signaling during transportation.’

In the Significance Statement, the phrase now reads ‘These alterations are likely due to the mechanical stress and dysfunction of critical neuromodulators, which compromise tissue integrity and neuronal function.’

In Results, page 15, it is stated now: ‘In summary, based on the experiments with NE, we conclude that altered neuromodulator signaling during slice transport and handling likely contributes to the reduction of LRDs.’

- c. Is there a quantification or visualization backing this statement: “This induces LRD-like synaptic activity in neurons that did not exhibit LRD-like activity in artificial CSF.”

Response: Since the quantification is not possible at this stage due to the low number of cases, we moved this statement from the Results to the Discussion, see below:

In Discussion, page 18: ‘To further investigate whether altered neuromodulator signalling contributes to the reduction of LRD observed in off-site recorded neurons, a small set of experiments was performed by perfusing the slices with human CSF. The preliminary data showed that the human CSF was able to induce LRD-like synaptic activity in human cortical neurons that did not exhibit LRDs in artificial CSF.’

Minor:

1. The same statistical approach used for the pyramidal cells should also be applied when comparing the FS interneurons. If the data on paired surgery level or transport only are not available, then at least the LME results should be reported and the lack of other controls discussed.

Response: We addressed this point in the manuscript now, see below:

In Results, on page 13, a paragraph was added:

‘For consistency, we also used an LME model to analyze the intrinsic properties of L2/3 FS interneurons. The analysis revealed that only 2 out of 12 parameters showed statistically significant differences between on-site and off-site: AP half-width ($p = 5.5 \times 10^{-4}$) and AP amplitude ($p = 3.3 \times 10^{-5}$). While other parameters, e.g., the resting membrane potential ($p = 0.32$), input resistance ($p = 0.059$), time constant ($p = 0.23$), Sag ($p = 0.11$), rheobase current ($p = 0.41$), AP threshold ($p = 0.47$), AP rise time ($p = 0.065$), AHP

amplitude ($p = 0.18$), maximum firing frequency ($p = 0.71$) and F-I slope ($p = 0.51$), showed no significant differences. Compared to the statistical results obtained with the non-parametric Wilcoxon-Mann-Whitney two-sample rank test (see Fig. 5 and Table 3), the analysis using the LME model showed very similar results. As data at the paired surgery level or for transport-only conditions were unavailable for FS interneurons, no additional analyses were conducted for these scenarios.'

2. The statement that ICC values of 0.3 or 0.16 indicate that differences in V_{rest} and FI-slope were "likely due to the surgeries" is somewhat misleading. These ICCs simply reflect how much variance is attributable to surgery versus recording location, not that surgery is the primary cause of the observed differences.

Response: We changed the sentences as follows:

In Results, on page 11: 'Only V_{rest} and F-I slope showed a meaningful between-surgery variability (intraclass correlation coefficient, ICC = 0.293 and 0.159, respectively). This indicates that the surgical cases contributed to the overall variance in these two parameters.'

The sentence 'Therefore, differences revealed in V_{rest} and F-I slope were likely due to the surgeries' was deleted.

3. "The findings of this study highlight how transport-induced stress can lead to significant alterations in synaptic efficacy and network dynamics, notably the reduction in EPSP amplitudes and LRDs." Please specify "reduction in EPSP amplitudes of FS interneurons"

Response: Done.

4. The lack of difference in RMP in FS interneurons is interesting, since this was so striking in the pyramidal cell dataset. Could this be mentioned and potentially discussed?

Response: We addressed this point in the revised manuscript, see below:

In Results, on page 12, it now reads: 'Although the resting membrane potential was comparable between FS interneurons recorded on- and off-site, the AP half-width was narrower in interneurons recorded off-site compared to those recorded on-site, indicating a change in the temporal dynamics of action potential generation (see Table 3).'

In Discussion, on page 17, it now reads: 'For FS interneurons, while there was no change in the resting membrane potential, off-site recordings showed a narrower action potential (AP) half-width and an increased amplitude compared to on-site recordings.'

5. Fig 7 / LRD: It appears that the authors used a one-tailed chi-square test. Since chi-square tests do not generally test a directional hypothesis and given the small number of observations, a two-tailed Fisher's exact test would be more appropriate. The results of the chi-square test or Fisher's exact test should also be referenced in the manuscript.

Response: We changed the manuscript accordingly, see below:

In Results, on pages 13,14, in the section 'Comparison of LRD occurrence in brain slice samples recorded on-site and off-site':

'There was a significant difference in the percentage of on- and off-site recorded neurons displaying sLRDs ($p = 0.043$ for Chi-square test without Yates correction).' and 'Hence, the

incidence of LRDs was significantly reduced in the recordings at the off-site laboratory (off-site 7% vs. on-site 21%; $p = 0.031$ for Chi-square test without Yates correction),’.

6. If n numbers are provided within the abstract (n = 200 neurons from 32 surgeries), then it should also or instead state the number of neurons that passed quality control and went into analyses.

Response: We added the number in the abstract, see below:

In the Abstract: ‘(n = 200 neurons from 32 surgeries, in which 112 neurons passed quality control for further analyses).’

7. The discussion reads: “The mechanical stress encountered during transport **can** alter these ECM components, leading to changes in tissue hydration, viscoelastic properties, and membrane fluidity, ultimately affecting cellular behaviors such as signaling pathways.” As this is a speculation, I would suggest rephrasing “can” to “could”.

Response: Done.

Dear Dr Koch,

Re: JP-RP-2025-288111R2 "Transport-Related Effects on Intrinsic and Synaptic Properties of Human Cortical Neurons: A Comparative Study" by Guanxiao Qi, Danqing Yang, Aniella Vanessa Bak, Werner Hucko, Daniel Delev, Hussam Aldin Hamou, Dirk Feldmeyer, and Henner Koch

We are pleased to tell you that your paper has been accepted for publication in The Journal of Physiology.

Yours sincerely,

Katalin Toth
Senior Editor
The Journal of Physiology

IMPORTANT POINTS TO NOTE FOLLOWING ACCEPTANCE OF YOUR PAPER:

- **IMPORTANT NOTICE ABOUT OPEN ACCESS:** To assist authors whose funding agencies mandate immediate public access to published research findings, The Journal of Physiology allows authors to pay an Open Access (OA) fee to have their papers made freely available immediately on publication.

- You can help your research get the attention it deserves! Check out Wiley's free Promotion Guide for best-practice recommendations for promoting your work at: www.wileyauthors.com/eoo/guide. You can learn more about Wiley Editing Services which offers professional video, design, and writing services to create shareable video abstracts, infographics, conference posters, lay summaries, and research news stories for your research at: www.wileyauthors.com/eoo/promotion.

- If you would like to receive our 'Research Roundup', a monthly newsletter highlighting the cutting-edge research published in The Physiological Society's family of journals (The Journal of Physiology, Experimental Physiology, Physiological Reports, The Journal of Nutritional Physiology and The Journal of Precision Medicine: Health and Disease), please click this link, fill in your name and email address and select 'Research Roundup': <https://www.physoc.org/journals-and-media/membernews>